# Constitutive activation of NF-κB inducing kinase (NIK) in the mesenchymal lineage using Osterix (Sp7)- or Fibroblast-specific protein 1 (S100a4)-Cre drives spontaneous soft tissue sarcoma

Jennifer L. Davis[1,2], Roman Thaler[3,4], Linda Cox[1,2], Biancamaria Ricci[1,5], Heather M. Zannit[1,5], Fei Wan[6], Roberta Faccio[1,5,7], Amel Dudakovic[3,4], Andre J. van Wijnen[3,4], Deborah J. Veis[1,2,7] *

1 Musculoskeletal Research Center, Washington University School of Medicine, St. Louis, MO, United States of America, 2 Department of Medicine, Division of Bone and Mineral Diseases, Washington University School of Medicine, St. Louis, MO, United States of America, 3 Department of Orthopedic Surgery, Mayo Clinic, Rochester, MN, United States of America, 4 Department of Biochemistry and Molecular Biology, Mayo Clinic, Rochester, MN, United States of America, 5 Department of Orthopaedic Surgery, Washington University School of Medicine, St Louis, MO, United States of America, 6 Department of Surgery, Division of Public Health Sciences, Washington University School of Medicine, St. Louis, MO, United States of America, 7 Shriners Hospitals for Children–St. Louis, St. Louis, MO, United States of America

* dveis@wustl.edu

**Data Availability Statement:** All relevant data are contained within the manuscript and its Supporting

## Abstract

Aberrant NF-κB signaling fuels tumor growth in multiple human cancer types including both hematologic and solid malignancies. Chronic elevated alternative NF-κB signaling can be modeled in transgenic mice upon activation of a conditional NF-κB-inducing kinase (*NIK*) allele lacking the regulatory *TRAF3* binding domain (*NT3*). Here, we report that expression of *NT3* in the mesenchymal lineage with *Osterix (Osx/Sp7)-Cre* or *Fibroblast-Specific Protein 1 (FSP1)-Cre* caused subcutaneous, soft tissue tumors. These tumors displayed significantly shorter latency and a greater multiple incidence rate in *Fsp1-Cre;NT3* compared to *Osx-Cre;NT3* mice, regardless of sex. Histological assessment revealed poorly differentiated solid tumors with some spindled patterns, as well as robust RelB immunostaining, confirming activation of alternative NF-κB. Even though *NT3* expression also occurs in the osteolineage in *Osx-Cre;NT3* mice, we observed no bony lesions. The staining profiles and pattern of *Cre* expression in the two lines pointed to a mesenchymal tumor origin. Immunohistochemistry revealed that these tumors stain strongly for alpha-smooth muscle actin (αSMA), although vimentin staining was uniform only in *Osx-Cre;NT3* tumors. Negative CD45 and S100 immunostains precluded hematopoietic and melanocytic origins, respectively, while positive staining for cytokeratin 19 (CK19), typically associated with epithelia, was found in subpopulations of both tumors. Principal component, differential expression, and gene ontology analyses revealed that *NT3* tumors are distinct from normal mesenchymal tissues and are enriched for NF-κB related biological processes. We conclude that

Information files. RNA-Seq data have been deposited in NCBI's Gene Expression Omnibus (https://www.ncbi.nlm.nih.gov/geo/) under the GEO Series accession number GSE158479.

**Funding:** This work was supported by NIAMS, National Institutes of Health Grants R01 AR052705 (to DJV), R01 AR070030 (to DJV and RF) and R01 AR066551 (to RF), R01 AR049069 (to AJvW), F31 AR068853 (to JLD), T32 AR060719 (to HZ); and NCI, National Institutes of Health, grants R01 CA235096 (to RF), and T32 CA113275 (to JLD). Histology services were provided by the Washington University Musculoskeletal Research Center Histology and Morphometry Core supported by P30 AR074992 and the Northwestern University Research Histology and Phenotyping Laboratory supported by P30 CA060553 awarded to the Robert H Lurie Comprehensive Cancer Center. Slide images were captured in the Alafi Neuroimaging Laboratory supported by S10RR027552. Additional support was provided by the Siteman Cancer Center Investment Program, St. Louis, MO to RF. Funding from Shriners Hospitals for Children was provided to DJV and RF. The funders had no role in study design, data collection and analysis, decision to publish, or preparation of the manuscript. The content is solely the responsibility of the authors and does not necessarily represent the official views of the National Institutes of Health.

**Competing interests:** The authors have declared that no competing interests exist.

constitutive activation of the alternative NF-κB pathway in the mesenchymal lineage drives spontaneous sarcoma and provides a novel mouse model for NF-κB related sarcomas.

# Introduction

NF-κB signaling is best known for its pivotal roles in innate and adaptive immunity [1]. However, this transcription factor family can also regulate both physiologic and pathologic processes outside of the immune system such as in bone and skin, and has been implicated in tumorigenesis [1–4]. The NF-κB pathway has two principal arms, classical and alternative. Under basal conditions, the five NF-κB subunits (RelA/p65, c-Rel, RelB, p50, and p52), which dimerize to form transcription factors, remain in the cytoplasm bound to inhibitors of κB (IκBs) rendering them inactive. A multitude of stimuli (including inflammation, pathogens, and DNA damage) can trigger a rapid classical NF-κB cascade which usually resolves in minutes to hours. Classical signaling is largely mediated through TAK1 phosphorylation of the IKKα/β/γ complex resulting in release of p65-containing dimers [5]. On the other hand, a much more limited repertoire of inputs feeds into the alternative NF-κB pathway (BAFFR, LtβR, CD40, RANKL). Induction of alternative NF-κB signaling requires de novo transcription and results in much slower kinetics on the order of hours to days. The alternative NF-κB pathway is triggered exclusively through NF-κB-inducing kinase (NIK) which phosphorylates IKKα to mobilize RelB/p52-containing dimers into the nucleus [1, 6]. Both *RelB* and *Nfkb2*, the gene encoding p52, have κB binding sites in their promoters and are transcriptionally upregulated by classical NF-κB stimuli. Importantly, overexpression and/or activating mutations in NIK can drive both classical and alternative signaling arms. Thus, while activation of either the classical or the alternative NF-κB program is highly stimulus-dependent, significant crosstalk exists between these two pathways [7].

Although direct mutations in upstream kinases or DNA-binding subunits of the NF-κB pathway are relatively rare, hyperactivation of NF-κB signaling is a common feature of a variety of cancers [8, 9]. Perturbations of the NF-κB pathway are prevalent in hematological malignancies, particularly B-cell lymphomas and multiple myeloma, but also drive many solid tumors (breast, liver, lung, prostate, and others) [10]. Moreover, elevated NF-κB expression is associated with worse overall survival in solid tumors [11]. NF-κB interacts with other signaling pathways such as mTOR, and with both transcriptional coactivators (CBP) as well as transcriptional corepressors (SMRT) to spur tumor growth [12]. NF-κB is also well-documented to antagonize the classical tumor suppressor p53 pathway in multiple cancer types [3]. Less commonly, NF-κB signaling is tumor suppressive as reflected by tumor formation in mice exhibiting loss of IKKγ in liver [13] and IKKα in skin [14]. Thus, dysregulation of NF-κB, in either direction, may promote tumor proliferation.

Compared to classical NF-κB, far less studies have been published on the alternative pathway, even though the pinnacle kinase of this signaling arm, NIK, is activated in multiple cancer types, including multiple myeloma, carcinomas (pancreas, ovarian and lung) and melanoma [15, 16]. Downstream of NIK, increased expression of the alternative NF-κB subunits RelB and/or p52 has been implicated in the pathogenesis of various other carcinomas and gliomas [17, 18]. Additionally, transcriptional activity of RelB/p52 can cooperate with genome stability enzymes, such as APOBECs and the TERT catalytic subunit of telomerase, to fuel tumor growth [19, 20]. Lastly, RelB/p52 can regulate the transcriptional suppressor EZH2 which inhibits transcription of well-known tumor suppressors (p16$^{Ink4a}$ and p14$^{ARF}$) and p53 target genes [17] in a variety of cancer types.

Alternative NF-κB pathway output is tightly controlled with multiple layers of negative feedback, all coalescing to maintain low basal NIK levels [6]. Normally, NIK is bound to TRAF2/3 and is continually K48-ubiquitinated by the E3 ligase, cIAP1/2, tagging it for proteasomal degradation. Upon signal induction, cIAP1/2 instead ubiquitinates TRAF3, leading to stabilization and accumulation of NIK. Removal of the TRAF3 binding domain of NIK stabilizes protein levels, leading to persistent elevation of alternative NF-κB signaling. Transgenic mice bearing a *NIK* allele lacking the *TRAF3* binding motif (*NT3*) downstream of a loxP/Neo-STOP/loxP cassette knocked into the *ROSA26* locus permit targeted expression of a constitutively active form of *NIK* with any *Cre*-recombinase driver [21]. Given the critical function of NIK in multiple myeloma [15], it was quite surprising that *NT3* mice crossed to *CD19-Cre* (pre B-cell) presented only with B-cell hyperplasia and not lymphoma or myeloma [21]. However, other studies using *CD19-Cre* show that insults to either Bcl6 [22] or Notch [23], in conjunction with the *NT3* transgene, did lead to oncogenic transformation. The *NT3* allele expressed in T-cells caused fatal inflammation and autoimmunity, but not lymphoma [24]. *NT3* expression early in the hematopoietic lineage also drove unchecked inflammation as well as bone marrow failure, but again, the pathogenesis in these mice arrested as a myeloid dysplastic syndrome rather than frank malignancy. Surprisingly, inducible global expression of *NT3* delayed the onset of acute myeloid leukemia driven by another oncogene [25]. In sum, *NT3* allele expression in these studies is necessary but not sufficient to promote tumor growth and may in some instances even suppress tumor formation.

Our laboratory has recently shown that constitutive activation of NIK in the osteolineage in *Osterix-Cre;NT3 (Osx-Cre;NT3)* mice enhances both basal and loading-induced bone formation [26]. Unexpectedly, these *Osx-Cre;NT3* mice also developed malignant subcutaneous, soft tissue tumors in early adulthood. To date, there have been very limited reports of either classical or alternative NF-κB involvement in sarcomas. Balkhi and colleagues found increased NF-κB activity across a panel of human sarcoma cell lines [27]. In another study, targeted sequencing of human follicular dendritic cell sarcomas revealed a loss of the NF-κB regulatory genes (NFKBIA, CYLD, and A20) in ~38% of samples [28]. While NIK has been shown to directly interact with the transmembrane K15 protein in viral-associated Kaposi sarcoma, downstream signaling output converged on the classical subunit p65 [29]. Finally, a hybrid approach utilizing both human and murine models of undifferentiated pleomorphic sarcoma found NF-κB to be the most transcriptionally active pathway in these tumors, through crosstalk with YAP1 signaling. Moreover, inhibition of p65 was shown to reduce tumor burden [29]. Importantly, the aforementioned studies linking NF-κB and sarcoma mainly focused on the role of the classical NF-κB pathway in promoting oncogenesis.

Here, we report that conditional activation of NIK in *Osx-Cre;NT3* mice, as well as in mice with alternative NF-κB expression broadly in the mesenchymal lineage, using *Fibroblast-specific protein 1-Cre* (*FSP1-Cre;NT3)*, promotes tumorigenesis. Histology, immunohistochemistry and RNA-Seq analyses confirmed mesenchymal derivation. To our knowledge, this is the first report of an alternative NF-κB activation driver in sarcoma.

## Materials and methods

### Mice

Mice were housed communally in a temperature-controlled, pathogen-free barrier facility, with 12-hour light/dark cycles, and had ad libitum access to fresh chow and water. Mice were observed daily by Division of Comparative Medicine (DCM) staff and 1–2 times a week by laboratory personnel. Tumor-bearing animals were euthanized at a humane endpoint before tumor size surpassed ethical limits (2cm). Any other health concerns were reported promptly

to the DCM veterinarian and all treatment or euthanasia recommendations followed. All protocols were approved by Institutional Animal Studies Committee at Washington University School of Medicine (ASC protocol #19–1059). Details addressing Animal Research: Reporting of *In Vivo* Experiments (ARRIVE) guidelines for study design, sample size, outcome measures, statistical methods, experimental animals, experimental procedures, and results are indicated in their respective sections. No animals were excluded from analyses. As only *Cre*-positive animals developed tumors, randomization and blinding by genotype was not possible in our studies.

The transgenic mouse lines *NIKΔT3 (NT3)* [21] on a C57Bl/6J background and *Osx1-GFP*:: *Cre* Tet-OFF (*Osx-Cre*) [30] on a mixed C57Bl/6J and CD1 background were as previously described. *FSP1-Cre (S100a4-Cre)* [31] mice on a FVB/n background were a gift from Dr. Gregory Longmore [32]. *Osx-Cre;TdT* mice were generated by crossing the following lines from The Jackson Laboratory, USA: B6.Cg-Tg(Sp7-tTA,tetO-EGFP/*Cre*)1Amc/J (catalog #006361) and B6.Cg-Gt(ROSA)26Sor^tm9(CAG-tdTomato)Hze/J (TdT) (catalog #007909) [33]. All parental lines were maintained separately due to strain differences. Homozygous *NT3* females were crossed to heterozygous *Osx-Cre* males to generate *Osx-Cre;NT3* mice or to heterozygous *FSP1-Cre* males to generate *FSP1-Cre;NT3* mice. *Control* (*Ctrl*) animals are the *Cre*-negative littermates from each crossing. Genotyping was performed on tail DNA using a REDExtract-N-Amp Tissue PCR kit (XNAT, Millipore Sigma, USA). Primer sequences and run conditions are listed in S1 Table. *Osx-Cre;NT3* or Osx-*Cre*;TdT and their *Ctrl* littermates were fed a 200ppm doxycycline chow (Purina Test Diet #1816332–203, St. Louis, MO, USA) until weaning (P21-P22) then switched to standard chow to induce *NT3 or Tdt* transgene expression (Purina 5058, St. Louis, MO, USA).

## Tumor latency and multiple tumor incidence

Mice were checked 1–2 times weekly for the presence of at least one palpable tumor mass and both the age and number of tumor masses per animal at this time was recorded. Multiple tumor incidence was defined as the presence of $\geq$2 tumors in the same animal. For a few animals, (n = 1 *Osx-Cre;NT3* female and n = 3 *FSP1-Cre;NT3* males) only the presence/absence of tumors was recorded, but not location or number, so these were included only in tumor latency data. No tumors were observed in *Ctrl* animals from either strain for ages up to 1 year. Kaplan-Meier curves were used to calculate tumor latency.

## Immunohistochemistry

Tumors were excised and fixed in 10% neutral buffered formalin. Paraffin-embedding and sectioning to 4-5μm thickness was performed by the Musculoskeletal Research Center Histology and Morphometry core at Washington University. Immunohistochemistry staining was completed by the Mouse Histology and Phenotyping Laboratory at Northwestern University. Either a no primary negative (with goat primary) or IgG isotype negative (rabbit and rat primaries) was used to confirm positive signal in stained sections. The TROMA-III hybridoma product (CK19) developed by Kemler, R. was obtained from the Developmental Studies Hybridoma Bank, created by the NICHD of the NIH and maintained at The University of Iowa, Department of Biology, Iowa City, IA 52242. Antibodies are listed in S2 Table. n = 3 for each genotype, mixed male and female. Sex and location of tumors for images shown: H&E— *Osx-Cre;NT3* = female trunk; *FSP1-Cre;NT3* = female trunk, GFP—*Osx-Cre;NT3* = female perineal; *FSP1-Cre;NT3* = female perineal, RelB—*Osx-Cre;NT3* = female perineal; *FSP1-Cre;NT3* = female perineal, αSMA—*Osx-Cre;NT3* = male facial, *FSP1-Cre;NT3* = female perineal; vimentin—*Osx-Cre;NT3* = female perineal, *FSP1-Cre;NT3* = female perineal; CK19—*Osx-Cre;*

*NT3* = female facial, *FSP1-Cre;NT3* = female perineal; CD45—*Osx-Cre;NT3* = female perineal, *FSP1-Cre;NT3* = female perineal; S100—*Osx-Cre;NT3* = male facial, *FSP1-Cre;NT3* = female perineal. Whole slide images were acquired using standard brightfield settings on a NanoZoomer 2.0 HT whole- slide scanner (Hamamatsu Photonics, Hamamatsu City, Shizuoka, Japan). Negative control stains on additional tumor sections are provided in S1 Fig. Sections were evaluated by Dr. Deborah J. Veis, a board-certified anatomic pathologist.

## Genomic DNA recombination

Tumor, liver, and dorsal skin tissues were either stored in RNAlater (R0901, Millipore Sigma, USA) at -20C or flash frozen in liquid nitrogen and stored at -80C˚. Sex and location of tumors: mixed male and female, 2 facial and 1 perineal for *Osx-Cre;NT3* or 2 perineal and 1 trunk for *FSP1-Cre;NT3*. gDNA was extracted from tissue samples as previously described [26]. PCR cycling conditions using GoTaq polymerase (M7123, Promega, USA) are listed in S3 Table. GelRed (41003, Biotium) was added at a concentration of 1:10,000 to agarose gels for DNA visualization as well as a 1Kb Plus DNA Ladder (10787018, Invitrogen, USA) for band size determination. Gel images were captured using a ChemiGenius 2 system using an ethidium bromide filter with accompanying GeneSnap v7.12.02 software (Syngene, United Kingdom).

## RNA-Seq

Tumors were excised to remove any gross overlaying tissue, and either stored at -20C˚ in RNAlater (R0901; Sigma, USA) or flash frozen in liquid nitrogen, and stored at -80C˚. Frozen tumors were pulverized in liquid nitrogen followed by RNA extraction as previously described [26]. Samples were screened by quantitative real-time PCR (qPCR) for blood- (*Hba1*, *Hba2*, *and Hbb*) and monocyte-specific (*CD68*) markers to assess low peripheral blood content. Gene expression was normalized to the levels of *Gapdh*, a housekeeping gene. qPCR primer sequences are listed in S4 Table. A Nanodrop device (ND-2000; ThermoFisher, USA) and other RNA quality assessments (Bioanalyzer 2100; Agilent Technologies, USA) were used to determine RNA concentration and RNA integrity number (RIN score). Samples with sufficient RNA yields as well as RIN scores and DV200 scores were prioritized for RNA-Seq analysis as previously described [34, 35]. Final samples for analysis (n = 3 each genotype, all female, *Osx-Cre;NT3* = 3 trunk or *FSP1-Cre;NT3* = 1 trunk and 2 perineal) had an average RIN score of 7.7 for *Osx-Cre;NT3* and 8.5 for *FSP1-Cre;NT3*.

RNA sequencing was performed at the Advanced Genomics Technology Center (Mayo Clinic, Rochester, MN) using an Illumina HiSeq2500 sequencer as previously described [34, 35] with the following modifications. Raw reads were trimmed and aligned to the mouse genome mm10 with Histat2 (Galaxy Version 2.1.0) using standard settings. Normalized reads per gene (read count) and statistical analyses were performed using standard settings of featureCounts [36], a read summarization program which counts, quantifies, and assigns sequencing reads for specific genomic features like mRNAs. Differential expression of genes was calculated based on the above read counts using negative binomial generalized linear models in DESeq2 [37]. Publicly available datasets were downloaded from NCBI's Gene Expression Omnibus (GEO) (https://www.ncbi.nlm.nih.gov/geo/) [38] and checked for quality with FASTQC (Galaxy Version 0.72). *Osx-Cre;NT3* and *FSP1-Cre;NT3* tumors were grouped together as *NT3* tumors due to their high degree of similarity on principal component analysis. Normalized reads per gene of *NT3* tumors (group 1) were compared to a panel of benign tissue datasets (group 2) including non-loaded cortical bone from *Ctrl* and *Osx-Cre;NT3* mice (GSE133212) [26], skeletal muscle (GSE83541) [39], white adipose tissue (GSE112999), mature

cartilage (GSE110051), vein (GSE128870), dermal fibroblast (GSE60569) [40], blood (GSE116630), and liver (GSE92364 [41], GSE93380 [42]). Individual sample files used for each tissue type are listed in S5 Table. RNA-Seq data for the *NT3* tumors have been deposited in GEO under the accession number GSE158479. Differentially expressed genes were defined as having fold changes (FC) >1.5 and a Benjamini-Hochberg adjusted p-value < 0.05 between group 1 and group 2 (S6 Table).

### Gene ontology (GO) analysis

The 1,314 DEGs listed in S6 Table were mapped to the *Mus musculus* reference genome (22,265 reference IDs) using the PANTHER Overrepresentation Test (Released 20200407) for GO biological process complete (Fisher's exact test with FDR, p<0.05). Annotation version and release date: GO Ontology database DOI: 10.5281/zenodo.3727280 Released 2020-03-23 (http://geneontology.org/). The final query for *NT3* tumors included 1,022 unique IDs. Significant biological processes (Fisher's exact test FDR adjusted p-value $< 1e^{-3}$) as well as unmapped IDs (Riken clones, GM pseudogenes, and miRNAs) and IDs mapping to multiple genes are listed in S7 Table.

### Statistical analysis

Except for RNA-Seq analyses, statistics were computed using either GraphPad Prism v8 software (GraphPad Software, Inc., USA) or SAS v9.4 (SAS institute Inc., USA). Normalized reads per gene were $\log_{10}$-transformed and compared by one-way ANOVA followed by Dunnett's multiple comparisons test in Fig 5. For the continuous outcome of average number of tumors per mouse, a two-way ANOVA was performed including strain, sex, and strain by sex interaction followed by Sidak's multiple comparisons test. A Fisher's exact test was used to compare differences in single vs multiple tumor incidence within and between strains and the false discovery rate (FDR) adjustment was used to adjust p-values for the number of comparisons.

Kaplan-Meier curves for tumor latency were compared either within or between strains using a standard log-rank (Mantel-Cox) test if the proportional hazard assumption was reasonable. If the assumption was violated, an extended Cox proportional hazards model was applied. For *Osx-Cre;NT3* male and female mice, hazard ratios were calculated in two different time intervals (t<125 days or t≥125 days) by including two Heaviside functions (hv1 = 1*male if t<125, hv1 = 0 if t≥125; hv2 = 1*male if t≥125; hv2 = 0 if t<125) with assumption that the hazard ratios are constant in each time interval. *Ctrl* animals for each strain were considered censored at the end of the follow-up period. Values of p<0.05 were considered significant, and data are presented as mean ± SD. Sample sizes are indicated in the respective figure legends or methods section.

## Results

### *NT3* transgenic mice develop spontaneous soft tissue tumors in the mesenchymal lineage

In the course of characterizing the bone phenotype of the *Osx-Cre;NT3* mice [26], we observed spontaneous soft tissue tumors. Based on their subcutaneous location and histological appearance, we considered that a mesenchymal origin was likely, and therefore crossed the *NT3* transgenic mice with a *FSP1-Cre* (*S100a4*) line [31], which predominantly shows expression in the fibroblast population of multiple organs but has also been documented in the myeloid lineage [31, 32, 43–45]. Beginning at 6 weeks of age, we monitored these mice for palpable tumor masses 1–2 times per week.

Both *Osx-Cre;NT3* and *FSP1-Cre;NT3* mice presented with spontaneous, superficial soft tissue tumors with 100% penetrance (Fig 1A). *Ctrl*, *Cre*-negative littermates of either line, did not develop tumors during a 1-year follow-up period. Tumors from both transgenic lines were generally firm, beige, and nodular in appearance. *Osx-Cre;NT3* mice had masses that were typically larger, more cystic, and predominantly (95% of males, 68% of females) found on or near the face in the submandibular region. In addition to the face, *Osx-Cre;NT3* animals also presented with tumors on the trunk (15% of males, 50% of females), perineal area (5% of males, 26% of females), and rarely on the limbs (1 female, 3%) (S2A Fig). *FSP1-Cre;NT3* tumors often became ulcerated and were most commonly found in the perineal region (78% of males, 100% of females). Tumor locations for *FSP1-Cre;NT3* animals also included the face (78% of males, 58% of females), localized mainly anterior on the snout, as well as the trunk (11% of males, 67% of females) and on the limbs (11% of males, 25% of females) (S2A Fig). Consistent with the broader expression pattern of FSP1 in the mesenchymal lineage, we observed more tumors on average ($3.57 \pm 1.83$ vs $1.95 \pm 2.02$, **$p < 0.0018$) in *FSP1-Cre;NT3* animals, in addition to an earlier onset (79 vs 129 days, ****$p < 0.0001$) versus their *Osx-Cre;NT3* counterparts. Lastly, radiographic imaging confirmed no overt bony tumors in either line that would be indicative of an osteosarcoma (S2B Fig).

We next examined if there was a sex difference between the two mesenchymal *Cre*-driven *NT3* transgenic lines. There was no interaction between sex and strain for this between-group comparison. However, the percentage of mice presenting with multiple tumors is significantly higher with *FSP1-Cre* in both sexes (Fig 1B). The absolute number of tumors per mouse revealed a similar upward trend in either sex in *Osx-Cre;NT3* vs *FSP1-Cre;NT3*, but this was only statistically significant in female mice (Fig 1C). The lack of statistical significance in male animals in Fig 1C is likely due to the difference in sample sizes as there were nearly twice as many animals in the *Osx-Cre;NT3* vs the *FSP1-Cre;NT3* cohort.

Analysis of the sex by time interaction indicated a time-varying hazard ratio between sexes within the same strain only for *Osx-Cre;NT3* mice (*$p = 0.0173$). A Cox proportional hazards model in two different time intervals (t<125 days or t≥125 days) was applied to test for a possible divergence in tumor latency by sex. In the first time interval (t<125 days), *Osx-Cre;NT3* males had a longer tumor latency than females, but this difference was not statistically significant. However, in the second time interval (t≥125 days), *Osx-Cre;NT3* males had a considerably shorter tumor latency period than females (Fig 1A and 1D). A similar tumor latency was observed between *Osx-Cre;NT3* males with single vs multiple tumors (129 vs 126 days) (S2C Fig). *Osx-Cre;NT3* females that presented with multiple tumors had a markedly longer median tumor latency period than females with only single tumors (161.5 days vs 112 days) (S2C Fig). Notably, neither average number of tumors, multiple tumor incidence, nor tumor latency differed significantly between male and female *FSP1-Cre;NT3* mice (Fig 1D, S2C Fig). Overall, despite a mild statistical difference between male and female *Osx-Cre;NT3* mice in tumor latency, we concluded that the *NT3* transgene was similarly oncogenic in both sexes of either strain, and the primary differences between the strains were the shorter latency and higher number of tumors with *FSP1-Cre*.

## Spontaneous tumors derive from *NT3* transgene expressing cells

To verify *NT3* transgene expression in tumor tissue, we first performed PCR on gDNA from whole tumors. Primers were designed to flank the loxP/Neo-STOP/loxP cassette [21]. In the absence of *Cre*-mediated recombination, the intact allele results in a larger band, inefficiently amplified in these PCR conditions (Fig 2A, *NT3* Tg STOP). If the loxP/Neo-STOP/loxP cassette is excised, a smaller band is present (Fig 2A, *NT3* Tg GO). Tumors from both *Osx-Cre;*

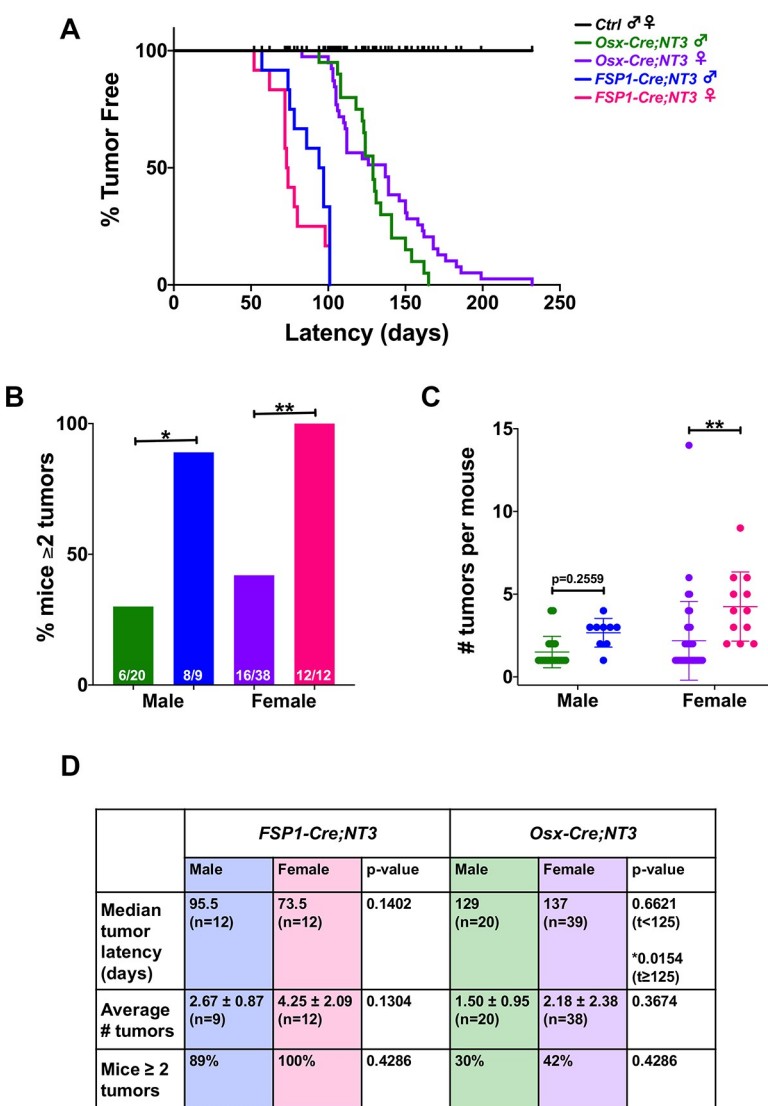

**Fig 1. *NT3* transgenic mice develop spontaneous soft tissue tumors. (A)** Kaplan-Meier curves showing tumor latency by sex in each strain. **(B)** Comparison between strains of males or females for the percentage of mice with $\geq 2$ tumors or **(C)** number of tumors per mouse. **(D)** Comparison by sex of the median tumor latency, average number of tumors per mouse, and percentage of mice with $\geq 2$ tumors in *Osx-Cre;NT3* and *FSP1-Cre;NT3*. Sample sizes are as indicated. Results are presented as mean ± SD. Standard log-rank (Mantel-Cox) test or extended (time-varying) Cox proportional hazards model for Kaplan-Meier curves. Two-way ANOVA followed by Sidak's multiple comparisons test for numbers of tumors per mouse or Fisher's exact test with FDR adjustment for percentage of mice with $\geq 2$ tumors. $^{*}p < 0.05$, $^{**}p < 0.01$.

*NT3* (Fig 2A, lanes 4–6) and *FSP1-Cre;NT3* (Fig 2A, lanes 13–15) mice show robust recombination of the *NT3* allele. Liver samples from these same mice served as a negative control for the recombination event (Fig 2A, lanes 1–3; 10–12). A parallel PCR of the *Crh*, corticotropin releasing hormone, locus was used as a positive control for gDNA extraction of whole tissue preps (Fig 2A, bottom panel). Furthermore, as the location of these tumors was subcutaneous, we also checked for presence of the recombination band in whole skin tissue to see if the tumors were perhaps arising from *Cre* expression in dermal fibroblasts. The *NT3* Tg GO product was seen in skin from *FSP1-Cre;NT3* mice (Fig 2A, lanes 16–18) but not in skin from *Osx-*

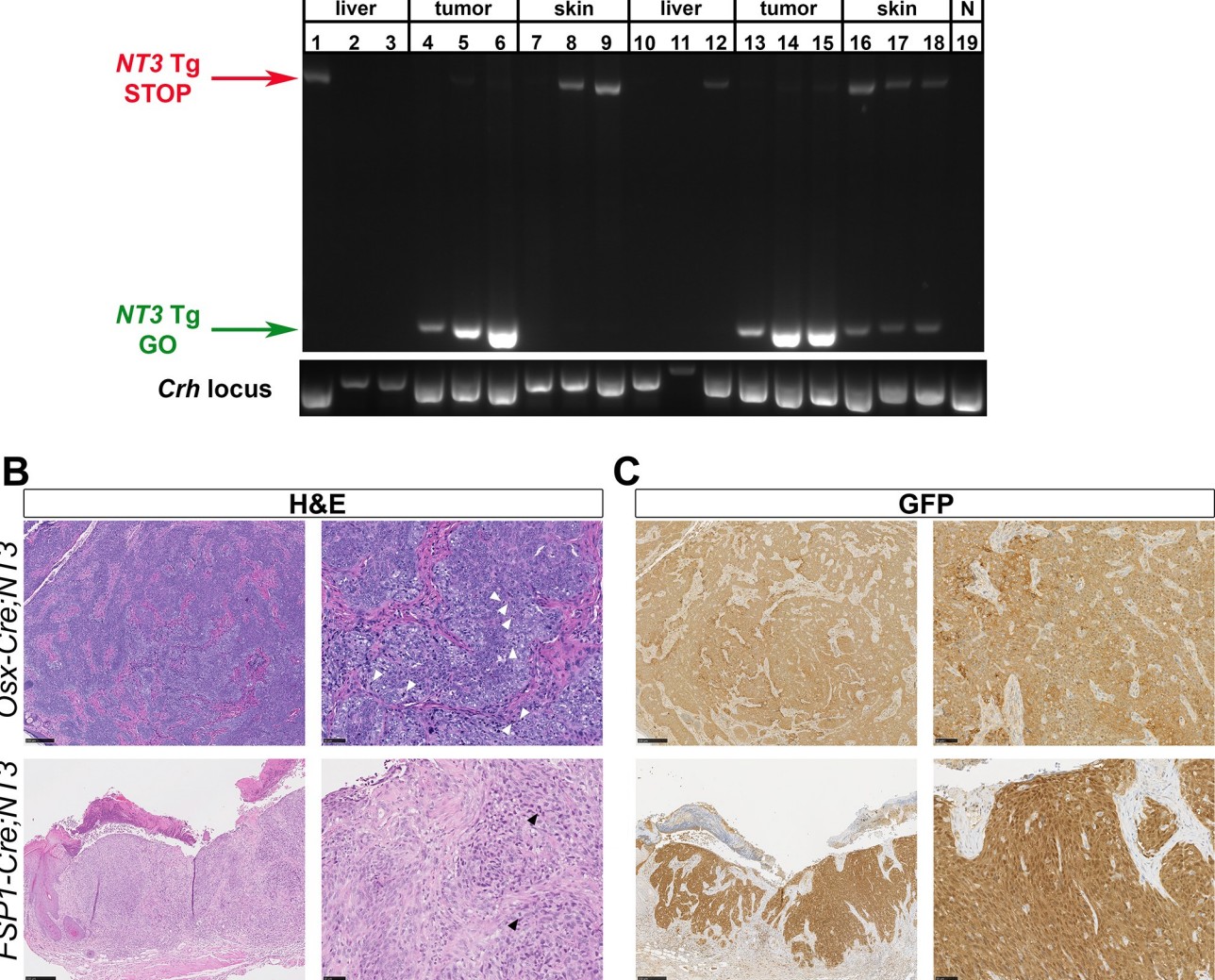

**Fig 2. Tumors derive from *NT3*-transgene expressing cells. (A)** gDNA PCR of whole tissue from liver, tumor, or skin in *Osx-Cre;NT3*, *FSP1-Cre;NT3*, or *Ctrl* animals. Amplification across the loxP/Neo-STOP/loxP cassette in the intact allele or after *Cre*-mediated recombination. *NT3* Tg = *NT3* transgene. N = *Cre*-negative, *Ctrl*. *Crh*, corticotropin releasing hormone. **(B)** Representative H&E staining or **(C)** immunohistochemistry for GFP which is expressed by the *NT3* transgene as well as the *Cre* construct in *Osx-Cre* mice. (*left images*) 10x, scale bars = 250μM and (*right images*) 40x magnification, scale bars = 50μM.

*Cre;NT3* (Fig 2A, lanes 7–9) or *Ctrl* animals (Fig 2A, lane 19), suggesting that *FSP1-Cre* drives significant recombination in benign skin. *FSP1-Cre* is known to be expressed in dermal fibroblasts [31, 43], but was not found in keratinocytes [32]. Similarly, skin from *Osx-Cre;Tdt* reporter mice did not show positive Tdt expression in keratinocytes (S3A and S3B Fig). Interestingly, we did observe a rare population of Tdt+ cells in the dermis as well as in cultured dermal fibroblasts from *Osx-Cre;Tdt* mice (S3 Fig). These findings suggest that *NT3* transgene recombination occurs in dermal fibroblasts of both strains, with the greater extent in *FSP1-Cre;NT3* mice corresponding to a greater number and earlier onset of tumors.

H&E staining revealed a similar histologic pattern in both *Osx-Cre;NT3* and *FSP1-Cre;NT3* tumors. The tumors have a solid and sometimes nested growth pattern, usually with some

areas of elongated, spindled cell shapes. Tumor cells are enlarged, with pleomorphic nuclei and frequent mitoses (Fig 2B). Some tumors have regional necrosis. Despite these aggressive features, tumor borders were largely pushing rather than infiltrative. As both the *NT3* transgene and the *Osx-Cre* driver express a GFP tag, we used an antibody to GFP to immunostain the tumors. The majority of tumor cells (75–100%) are GFP-positive in both *Osx-Cre;NT3* and *FSP1-Cre;NT3* tumor samples (Fig 2C). The immunostained sections also highlight nests of tumor cells at the dermal-epidermal junction in many *FSP1-Cre;NT3* tumors (Fig 2C, bottom right).

## *NT3* tumors share markers of soft tissue sarcomas

We next sought to determine the cellular origin of these spontaneous tumors using immuno-histochemistry (Fig 3). Use of either the *Osx-Cre* or *FSP1-Cre* drivers targets *NT3* transgene expression to the mesenchymal lineage. However, evidence for NF-κB, and particularly alternative NF-κB, in the pathogenesis of sarcomas is very limited and largely associative [46]. Nevertheless, we proceeded to investigate two well-known mesenchymal lineage markers, alpha-smooth muscle actin (αSMA) and vimentin (Vim), for which anti-mouse antibodies are available. Both *Osx-Cre;NT3* and *FSP1-Cre;NT3* tumors (n = 3 of each) exhibit moderate to strong αSMA staining in at least 60% of the tumor cells. Strong expression of Vim was seen in most (70–80%) tumor cells from *Osx-Cre;NT3* animals, but Vim staining was restricted to the periphery in *FSP1-Cre;NT3* tumors, where 20–50% of cells were weakly to moderately positive. Furthermore, staining of the epithelial marker, cytokeratin 19 (CK19), revealed a dichotomous expression pattern in these tumors with areas of positive as well as negative staining within the tumor mass. Interestingly, the CK19+ cells appeared morphologically distinct as large, spindle-shaped cells compared to the smaller, rounder tumor cells.

Both the *Osx-Cre* and *FSP1-Cre* drivers have been previously shown to have off-target effects outside of the mesenchymal lineage, most notably in cells of hematopoietic origin [32, 33, 44, 45, 47, 48]. To this end, we assayed CD45 expression. While tumor samples from both mesenchymal *Cre*-drivers showed positive CD45 staining, this appeared to be restricted to the inflammatory infiltrate. *FSP1-Cre;NT3* tumor samples exhibited a higher degree of this CD45 infiltrate, most likely owing their propensity to be superficially ulcerated at presentation.

As melanoma is almost always part of the diagnostic workup for a poorly differentiated sub-cutaneous tumor in patients, we checked for S100 expression [49]. Remarkably, both

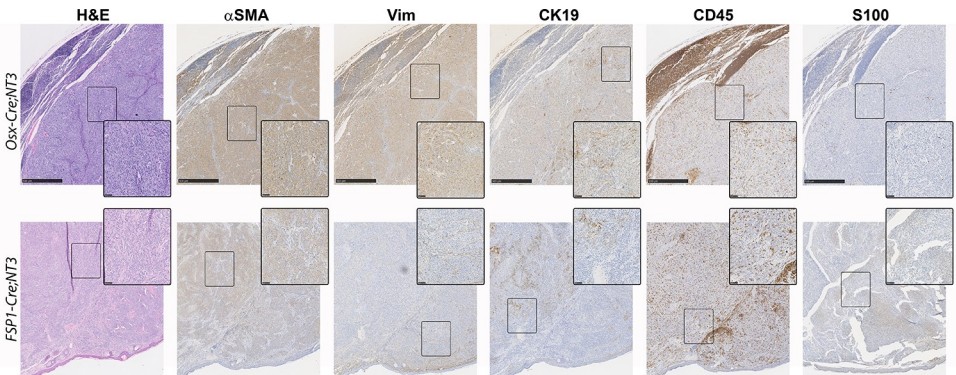

**Fig 3. *NT3* tumors have features of soft tissue sarcomas.** Immunohistochemical staining of representative *Osx-Cre; NT3* (top) and *FSP1-Cre;NT3* (bottom) tumors for hematoxylin and eosin (H&E), alpha-smooth muscle actin (αSMA), vimentin (Vim), Cytokeratin 19 (CK19), CD45, and S100 protein. Scale bars = 500μM. Insets are taken from boxed regions, with scale bars = 50μM.

*FSP1-Cre;NT3* and *Osx-Cre;NT3* tumors were negative for S100 staining, except for a few individual cells that are likely skin macrophages that have taken up melanosomes from benign melanocytes. In sum, the totality of the above morphological and immunohistochemistry evidence suggests that these spontaneous tumors represent a poorly differentiated, aggressive sarcoma.

## NF-κB /NF-κB -related biological processes are enriched in *NT3* tumors

To gain insight into both the possible cellular origin and pathogenesis of these spontaneous tumors, we performed RNA-Seq analysis of tumors from both *Osx-Cre;NT3* and *FSP1-Cre; NT3* mice. We performed principal component analysis (PCA) of tumors and a panel of benign tissues using publicly available datasets from NCBI's Gene Expression Omnibus (GEO) [26, 39–42] (S5 Table), including cortical bone samples from *Osx-Cre;NT3* and littermate *Ctrl.* We also included two tissues or cell types of non-mesenchymal origin (e.g. blood and liver) as biological outgroups for PCA and hierarchical clustering. As expected, tumors from both lines segregated away from both blood and liver samples (Fig 4A). Notably, the *Osx-Cre;NT3* tumors clustered more closely with the *FSP1-Cre;NT3* tumors than to cortical bone from *Osx-Cre;NT3* mice [26]. Both *Osx-Cre;NT3* and *FSP1-Cre;NT3* tumors did not significantly overlap with any other mesenchymal tissue assayed (muscle, vein, fat, cortical bone, dermal fibroblast) except for mature cartilage. Anatomically, these tumors were not found near any cartilaginous sites, and histologically they did not show indications of cartilage differentiation such as lacunar morphology or matrix production. Furthermore, expression of selected cartilage and muscle marker genes was low compared to their respective benign tissues (S4 Fig). Based on this analysis, we concluded that the *Osx-Cre;NT3* and *FSP1-Cre;NT3* tumor expression profiles most closely match one another, suggesting a distinct, but as yet unclear, developmental origin.

Due to their high degree of similarity, we grouped together *Osx-Cre;NT3* and *FSP1-Cre; NT3* tumors (*NT3* tumors) and compared them to all remaining tissues shown in Fig 4A to generate a list of differentially expressed genes (DEGs). We further filtered the DEG list to only those genes with a fold-change (FC) >1.5 (S6 Table) as NF-κB chiefly functions as an activating transcriptional program. Moreover, the vast majority of DEGs in our previous study comparing *Ctrl* bone to *Osx-Cre;NT3* bone showed that differences skewed strongly towards increased, and not decreased, target gene expression as a result of the *NT3* transgene [26]. Inspection of the DEG list did not yield any evidence of a single oncogenic driver and/or tumor suppressor. Importantly, we did not observe abnormal expression levels of oncogenes known to lead to NF-κB activation including *p53*, *KRAS*, *EGFR*, or *PI3K* [3]. However, this lack of evidence could be due to the absence of a true matched reference for our *NT3* tumors since their etiology remains unknown.

As individual genes are rarely biologically informative in isolation, we performed pathway analysis. Gene ontology (GO) analysis for biological process (BP) of the 1,314 DEG genes from above (S6 Table) revealed 509 GO BP terms, 170 of which were highly significant (FDR adjusted p-value <1e$^{-3}$; S7 Table). To distill out the key pathways driving spontaneous tumor formation in *Osx-Cre;NT3* and *FSP1-Cre;NT3* mice, we further refined our search to those processes with a fold-enrichment ≥ 3 relative to the expected incidence in the mouse genome (Fig 4B). The most overrepresented program, with a 6.22 fold-enrichment score, was the GO term for inflammatory response to antigenic stimulus (GO:000243) while the GO term for regulation of IKK/NF-κB signaling (GO:0043122) displayed a 3-fold enrichment. Of the 22 total GO BP shown in Fig 4B, 73% (16/22 –black bars) are known NF-κB/NF-κB related terms, predominated by inflammation and immune processes. Other NF-κB related GO terms of interest

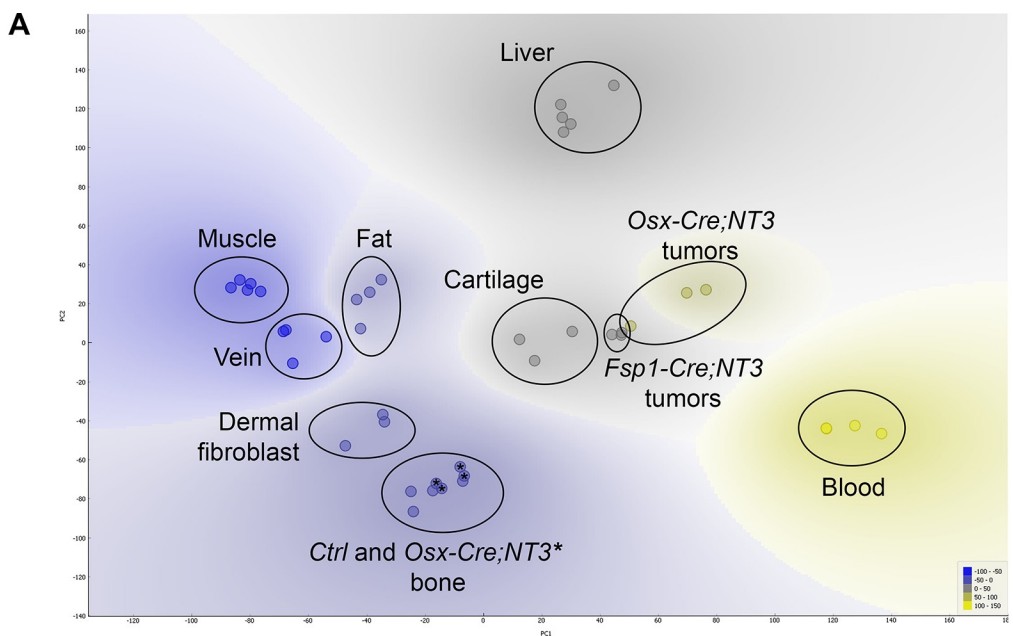

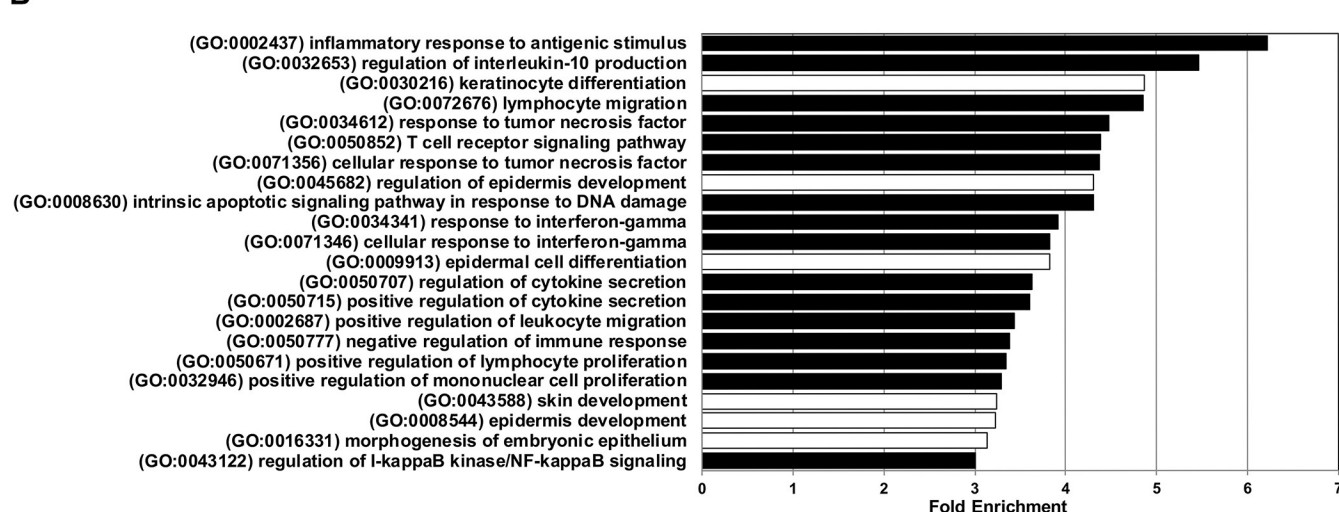

**Fig 4. *NT3* tumors cluster together and show upregulation of NF-κB biological processes. (A)** Principal component analysis of *Osx-Cre;NT3* tumors and *FSP1-Cre;NT3* tumors vs *Ctrl* bone and *Osx-Cre;NT3* bone (marked with an *) as well as other benign tissues—muscle, fat, vein, dermal fibroblast, mature cartilage, blood, liver. **(B)** Gene ontology (GO) analysis of *NT3* tumors vs the same panel of tissues in (A) showing the top biological processes (BP) with fold-enrichment ≥ 3 and a Fisher's exact FDR adjusted p-value < 1e$^{-3}$. black bars = NF-κB /NF-κB related processes; white bars = skin/epithelial processes.

include intrinsic apoptotic signaling pathway in response to DNA damage (GO:0008630) and both positive regulation of lymphocyte proliferation (GO:0050671) and positive regulation of mononuclear cell proliferation (GO:0032946), as these may indicate possible tumor cell proliferation. The remaining 27% GO BP (6/22 –white bars) represent skin development, particularly of the epidermis, which is likely contamination of overlaying skin incidental from gross tumor excision.

We next performed a gene set enrichment analysis (GSEA) focused on NF-κB related gene sets to determine if *NT3* tumors and *NT3* bone [26] would co-segregate, reflecting common transgene targets. However, with the large number of genes represented (~500 genes,

S8 Table), the tissue signature predominated, and all bones clustered with each other and away from the tumors (S5 Fig). To more specifically focus on *NT3* transgene targets, we turned to our previous comparison of gene expression in age/sex/tissue matched *Osx-Cre;NT3* and *Ctrl* bone. PCA of the top 20 DEGs and 3 NF-κB pathway genes that were upregulated in *Osx-Cre; NT3* vs *Ctrl* bones was performed, including all of the samples used in Fig 4A. All of the non-transgenic tissues including bone clustered together, and although the *NT3*-expressing tumors and NT3 bone were distinct from each other, they segregated away from this large cluster (Fig 5A). To further demonstrate NF-κB activation, we compared expression of the 3 NF-κB genes (Fig 5B), as well as several DEGs (Fig 5C), each with 2 or more κB response elements. Strikingly, the vast majority of these normal tissues had much lower levels of expression of all of these NF-κB targets, despite the varied tissues represented, and there were only rare cases in which individual tissues showed expression comparable to the NT3 tumors.

With this strong evidence of NF-κB upregulation in tumors, we turned to immunostaining to verify that the tumor cells themselves, rather than tumor stroma, show pathway activation. RelB is the principal NF-κB subunit downstream of NIK responsible for modulation of gene transcription. Strikingly, both *Osx-Cre;NT3* and *FSP1-Cre;NT3* mice show robust nuclear staining of RelB in all tumors (Fig 6A and 6B). Furthermore, there is a clear distinction between the uniformly RelB-positive tumor cells versus RelB-negative stromal cells. The majority of cells in adjacent benign tissues also lack nuclear RelB. Based on these gene expression and immunohistochemistry data, together with the presence of the *Cre*-mediated recombination product, in two distinct mouse lines, we concluded that *NT3*-initiated tumor formation is indeed driven by NF-κB activation.

## Discussion

Although the initial discovery of NF-κB arose from observations that the v-rel viral oncogene and translocations of c-rel can be oncogenic in lymphocytes, few mutations in genes of the NF-κB pathway have clear oncogenic effects [50]. In fact, the *NT3* transgene used here was first generated after multiple genetic aberrations in the alternative NF-κB pathway were identified in B cell malignancies. However, removal of the negative regulatory *TRAF3*-binding domain of *NIK* in this transgene was insufficient to cause oncogenesis in the B cell lineage [21–23]. Other studies have expressed the *NT3* transgene in hematopoietic stem cells, T cells, myeloid cells, intestine, pancreatic islets, and hepatocytes, without reports of malignant transformation [51–56]. NF-κB expression is found both within the epidermal and dermal layers of the skin with expression in the dermis restricted to dermal fibroblasts [57]. Intact NF-κB signaling, particularly upstream kinase activation, is essential to maintain epidermal homeostasis and to prevent squamous cell carcinoma [14]. Therefore, the presentation of subcutaneous tumors in *Osx-Cre;NT3* mice was unexpected. Our initial motivation in generating these conditional transgenic mice was to understand the role of alternative NF-κB in osteogenic cells, where *Osx* is highly expressed, but we never observed any bone tumors. Based on the location and histologic appearance, we hypothesized that the cell of origin was likely mesenchymal and in the dermis or subcutaneous soft tissues, and we were able to replicate the tumor phenotype with a *Cre* driver with that expression pattern, *FSP1-Cre*. Together, these two mouse lines displaying spontaneous tumorigenesis indicate that the loss of TRAF3 binding in NIK can be oncogenic in certain cellular contexts.

We used immunohistochemistry to attempt to classify *NT3* tumors by cell of origin. Both *Osx-Cre* and *FSP1-Cre* driven *NT3* tumors expressed the marker αSMA in most malignant cells, while vimentin staining was present, although more variable, in the latter colony. It is not clear why vimentin was less consistently expressed than αSMA. Nevertheless, αSMA and GFP

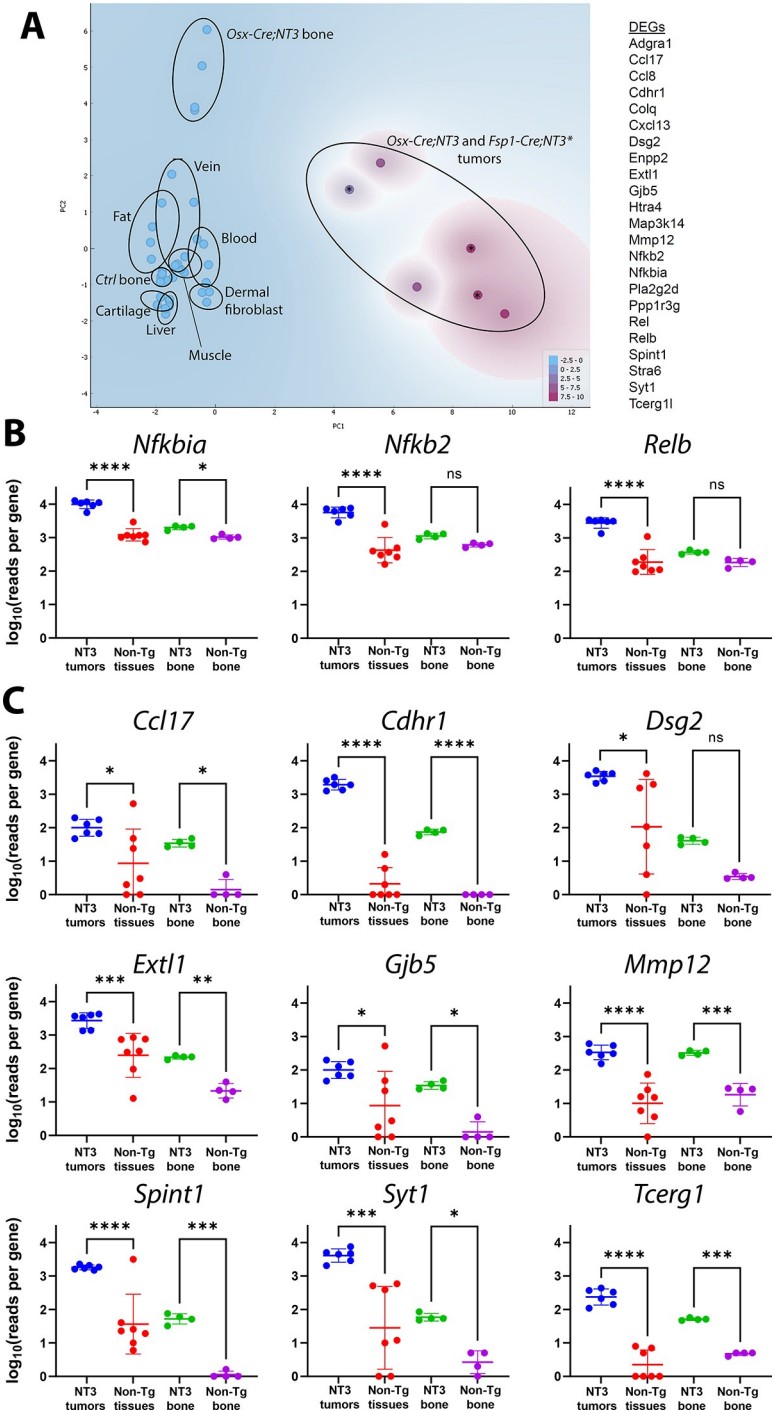

**Fig 5. Tumors show strong expression of NF-κB target genes. (A)** Principal component analysis of *Osx-Cre;NT3* tumors and *FSP1-Cre;NT3* tumors (marked with *), *Ctrl* and *Osx-Cre;NT3* bone, and other normal tissues utilizing the 23 genes listed. Expression of **(B)** NF-κB genes (*Nfkbia*, *Nfkb2*, and *Relb*) and **(C)** 9 additional *NT3* transgene targets previously identified as top DEGs in *NT3* bone vs *Ctrl* bone, each with 2 or more κB response elements. Log$_{10}$(reads per gene) is shown from normalized RNA-Seq data. One-way ANOVA, followed by Dunnett's multiple comparisons test; *, p<0.05, **p<0.01; ***, p<0.001; ****, p<0.0001.

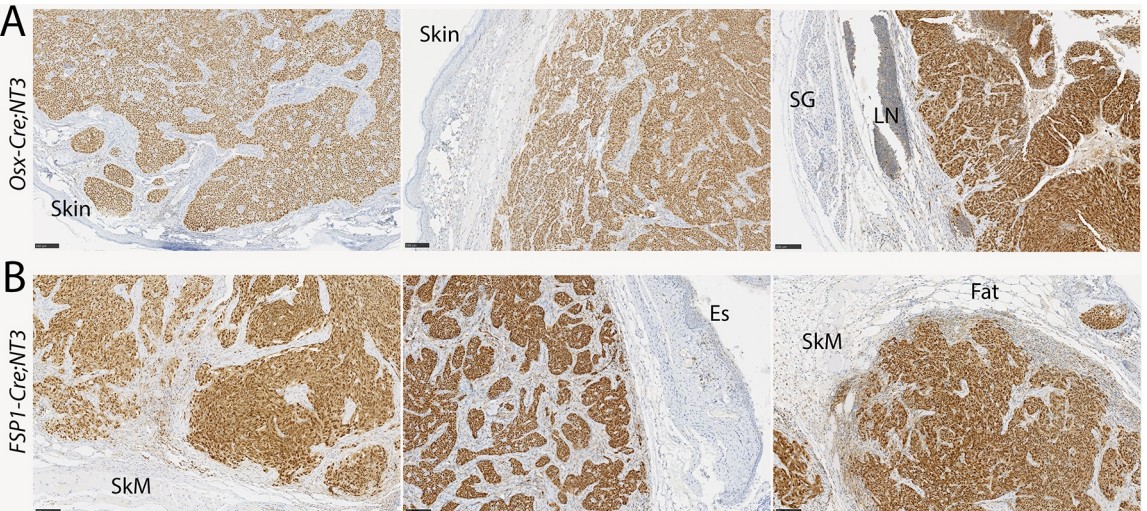

**Fig 6. The *NT3* transgene activates alternative NF-κB in tumor cells.** Immunostaining for RelB in **(A)** *Osx-Cre;NT3* or **(B)** *FSP1-Cre; NT3* tumors. RelB is uniformly nuclear in malignant cells, indicating robust activation of the alternative NF-κB pathway. Few cells in adjacent benign tissues show significant staining for RelB. SG, salivary gland; LN, lymph node; SkM, skeletal muscle; Es, esophagus. Scale bars = 100μM. n = 3 each genotype.

showed a very high degree of concordance. A mesenchymal origin is also aligned with that expected from the overlap in pattern of these *Cre* drivers. Thus, the totality of the evidence points towards a mesenchymal origin without differentiation towards a specific lineage. The classification of poorly differentiated sarcomas is notoriously difficult, with a paucity of diagnostic antibodies for mesenchymal populations, compounded here by limited cross-reactivity of antibodies to test tumors of murine rather than human origin. The presence of a significant population of CK19+ cells in both tumor types, in populations of large and pleomorphic cells indicative of tumor rather than host cells, is somewhat confounding. CK19 expression has been described mainly in thyroid, gastroenteropancreatic and hepatobiliary tumors [58, 59]. Because of the unique combination of location, overall histology, and expression patterns of the two *Cre* drivers, we consider an epithelial origin unlikely. The calcium binding protein S100, a marker that is highly expressed in melanomas, was completely absent in our tumors, and this result argues against a melanocytic derivation. Although expression of both *Osx-Cre* and *FSP1-Cre* alleles has been observed in CD45+ cells [32, 33, 44, 45, 48], the absence of CD45 in the tumor cells indicates that a hematopoietic origin is unlikely. In further support of this conclusion, expression of this *NT3* transgene in various hematopoietic cells did not result in malignancy [51–53].

We next attempted further classification of *NT3* tumors using RNA-Seq data and PCA, which suggested that the tumors are distantly related to tissues with cartilage expression signatures. However, histologic features are not consistent with a chondrogenic origin, which also seems unlikely in dermal or subcutaneous mesenchyme, and cartilage marker genes are poorly expressed. The *NT3* transgene is expressed in the skin of both *FSP1-Cre;NT3* and *Osx-Cre;NT3* animals, but PCA analysis did not indicate shared features with normal dermal fibroblasts. As bulk RNA-Seq analysis of *NT3* tumors is unlikely to uncover significant similarity to a distinct subpopulation of dermal fibroblasts, this does not rule out a dermal origin for tumorigenesis. Undoubtedly, our RNA expression analysis is limited by the datasets used for comparison. Without clear hypotheses to indicate other tissues or cells to include, further comparisons are outside the scope of this study. Such work might include filtering the data to reduce the

keratinocyte signature, since the tumors do not appear to have an epidermal origin, as well as comparisons to a panel of murine mesenchymal and perhaps epithelial tumors. Finally, one intriguing possibility is a follicular dendritic cell sarcoma, which can arise from ubiquitous perivascular cells, and in humans can show aberrant NF-κB regulation and high expression of CXCL13, an alternative NF-κB driven gene [60].

Based on early studies employing overexpression, NIK was originally named for its ability to activate classical NF-κB signaling [61]. Although it was later shown to be dispensable for this function, various studies employing the constitutively activated NIK allele used here have shown induction of both classical and alternative NF-κB signaling [1]. RNA-Seq analysis of *NT3*-driven tumors demonstrates activation of κB-responsive genes including RelB, but it is not clear if tumorigenesis requires classical and/or alternative signaling. Therefore, additional studies utilizing mice deficient in downstream components specific to each pathway would be required to demonstrate whether one or both pathways are necessary for the observed tumorigenesis.

The variable time to emergence of the *NT3* tumors, as well as a relatively small average number of tumors per mouse, suggests that a second genetic hit is likely needed in addition to NIK activation for oncogenic transformation. Previous studies have shown that Bcl6 [22] and Notch [23] collaborate with the *NT3* transgene to generate tumors. Yet, pathway analysis of RNA-Seq data from *NT3* tumors compared to several benign tissues failed to identify other clear oncogenic signals, including pathways such as p53 that have previously been linked to sarcomagenesis [46, 62]. Our study remains limited by the lack of a corresponding benign comparator to the cell of origin. Furthermore, chromosomal translocations are also a common feature of sarcomas, and we have not queried for this form of genomic change in the *NT3* tumors.

Although we wish to emphasize that the *NT3* transgene drove tumors in both sexes, with two different *Cre* drivers, we did observe some differences between the groups. With regard to the *Cre* driver, tumors in *FSP1-Cre;NT3* mice of both sexes appeared with shorter latency and in greater numbers. This is most likely a direct effect of constitutive expression of the *FSP1-Cre* allele in dermal fibroblasts. *Cre* expression in *Osx-Cre;NT3* mice, on the other hand, was induced postnatally in our studies, and detected only in a rare population of dermal fibroblasts. Despite subtle variations by sex in tumor latency in the *Osx-Cre;NT3* mice, none were present in the *FSP1-Cre;NT3* line. Many *Osx-Cre;NT3* males in our colony were utilized for other experiments prior to the appearance of tumors [26] and, therefore, the number of animals between sexes in this strain was not balanced. We also had fewer *FSP1-Cre;NT3* mice of both sexes, as these were generated later. The discrepancy in group size may have influenced the outcome of the sex-by-strain statistical analysis. Nevertheless, despite the fact that we previously observed sex differences in bone phenotype with modulation of the alternative NF-κB pathway [63], there appears to be little to no biologically significant effect of sex on tumorigenesis suggesting that *NT3* tumors form independent of male or female sex steroids.

In summary, we describe here a new, 100% penetrant, spontaneous soft tissue tumor model in mice expressing an activated form of NIK, the apex kinase in the alternative NF-κB pathway. Tumors arose using two distinct *Cre*-driver strains with overlapping mesenchymal lineage expression, supporting our conclusion that this mutant NIK allele has oncogenic potential. While NF-κB activation is a feature of many types of hematopoietic and epithelial tumors, it has not previously been intimately associated with mesenchymal tumors. Our observation that this particular NIK allele seems to have oncogenic potential only in the latter cell ontogeny suggests that there is much we do not yet understand about the role of NF-κB in cancer, specifically in the sarcoma setting.

## Supporting information

**S1 Fig. IHC negative control staining. (A)** No primary antibody stain for anti-goat GFP-biotin antibody. **(B)** Rat IgG background staining for CK19 antibody. **(C)** Rabbit IgG background staining for αSMA, vimentin, CD45, and S100 protein antibodies. All negative control stains were performed on *Osx-Cre;NT3*-tumor sections. Top row for each panel is 10x and the bottom row for each panel is 40x magnification. Scale bars = 100μM.
(TIF)

**S2 Fig. Additional tumor information for *Osx-Cre;NT3* and *FSP1-Cre;NT3* mice. (A)** Number of *Osx-Cre;NT3* and *FSP1-Cre;NT3* mice presenting with a tumor mass at different anatomical locations. In the case of multiple tumors within the same animal, the animal was counted once in each category for 1 or more masses at a given location. **(B)** Representative radiographic images revealing no overt osseous tumors in either strain. scale bar = 3mm. **(C)** Median tumor latency for *Osx-Cre;NT3* and *FSP1-Cre;NT3* animals presenting with single or multiple ($\geq$2) tumors. Samples sizes are as indicated in each panel. Standard log-rank (Mantel-Cox) test: ****$p<0.0001$.
(TIF)

**S3 Fig. Rare population of Tdt+ dermal fibroblasts in *Osx-Cre;Tdt* mice. (A)** Direct fluorescence of Tdt with DAPI counterstain in frozen skin sections from Ctrl or **(B)** *Osx-Cre;Tdt* mice. Representative 20x images shown with white arrows denoting Tdt+ cells. Scale bars = 20μm. **(A'-B')** Higher magnification of boxed areas in A-B. Scale bars = 5μm. n = 2 each genotype. **(C)** Flow cytometry analysis for Tdt in cultured skin fibroblasts from Ctrl or Osx-Cre;Tdt mice. n = 3 each genotype. Unpaired one-tailed t-test with Welch's correction: *$p<0.05$.
(TIF)

**S4 Fig. Low expression of cartilage and muscle markers in *Osx-Cre;NT3* and *FSP1-Cre;NT3* tumors. (A)** Expression of cartilage markers or **(B)** muscle markers in *NT3* tumors versus the respective tissue. Log$_{10}$(reads per gene) is shown from normalized RNA-Seq data. Mann Whitney U test; *, p$<$0.05, **p$<$0.01.
(JPG)

**S5 Fig. Tissue signature dominates over *NT3* transgene signature in broad NF-kB gene list.** Principal component analysis of *Osx-Cre;NT3* tumors, *FSP1-Cre;NT3* tumors, *Osx-Cre;NT3* bone, and *Ctrl* bone using genes from a gene set enrichment analysis (GSEA) of top NF-κB related gene sets listed in S8 Table.
(PNG)

**S6 Fig. Raw gel image for Fig 2.**
(TIF)

**S1 Table. Genotyping conditions.**
(XLSX)

**S2 Table. List of antibodies.**
(XLSX)

**S3 Table. Genomic DNA recombination conditions.**
(XLSX)

**S4 Table. q-RT-PCR primers for peripheral blood and monocyte markers.**
(XLSX)

**S5 Table. GEO datasets for Osx-Cre;NT3 and FSP1-Cre;NT3 tumors and benign tissues.**
(XLSX)

**S6 Table. Mean normalized counts for DEG list between NT3 tumors and benign tissues.**
(XLSX)

**S7 Table. GO analysis for top DEGs.**
(XLSX)

**S8 Table. Mean normalized counts NF-κB gene sets from GSEA between NT3 tumors and NT3 bone.**
(XLSX)

**S1 File.**
(DOCX)

## Acknowledgments

We wish to thank Crystal Idleburg and Samantha Coleman for their histological expertise as well as Gary London for imaging support. Katherine Gruner at Northwestern University provided valuable technical assistance with IHC.

## Author Contributions

**Conceptualization:** Jennifer L. Davis, Roman Thaler, Roberta Faccio, Andre J. van Wijnen, Deborah J. Veis.

**Data curation:** Jennifer L. Davis, Roman Thaler, Amel Dudakovic, Andre J. van Wijnen, Deborah J. Veis.

**Formal analysis:** Jennifer L. Davis, Roman Thaler, Biancamaria Ricci, Fei Wan, Amel Dudakovic.

**Funding acquisition:** Jennifer L. Davis, Heather M. Zannit, Roberta Faccio, Andre J. van Wijnen, Deborah J. Veis.

**Investigation:** Jennifer L. Davis, Roman Thaler, Linda Cox, Biancamaria Ricci, Heather M. Zannit, Amel Dudakovic.

**Methodology:** Jennifer L. Davis, Roman Thaler, Amel Dudakovic, Andre J. van Wijnen, Deborah J. Veis.

**Project administration:** Jennifer L. Davis.

**Resources:** Roberta Faccio, Amel Dudakovic, Andre J. van Wijnen, Deborah J. Veis.

**Software:** Roman Thaler.

**Supervision:** Roberta Faccio, Andre J. van Wijnen, Deborah J. Veis.

**Validation:** Jennifer L. Davis, Roman Thaler, Amel Dudakovic, Andre J. van Wijnen, Deborah J. Veis.

**Visualization:** Jennifer L. Davis, Roman Thaler, Linda Cox, Biancamaria Ricci, Heather M. Zannit, Amel Dudakovic, Deborah J. Veis.

**Writing – original draft:** Jennifer L. Davis, Deborah J. Veis.

**Writing – review & editing:** Jennifer L. Davis, Roman Thaler, Roberta Faccio, Amel Dudakovic, Andre J. van Wijnen, Deborah J. Veis.

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
