## [Decision Letter · Decision Letter 0]

19 Mar 2021

PONE-D-20-37281

Constitutive Activation of NF-κB Inducing Kinase (NIK) in the Mesenchymal Lineage using Osterix (Sp7)- or Fibroblast-Specific Protein 1 (S100a4)-Cre Drives Spontaneous Soft Tissue Sarcoma

PLOS ONE

Dear Dr. Veis,

Thank you for submitting your manuscript to PLOS ONE. After careful consideration, we feel that it has merit but does not fully meet PLOS ONE’s publication criteria as it currently stands. Therefore, we invite you to submit a revised version of the manuscript that addresses the points raised during the review process.

We look forward to receiving your revised manuscript.

Kind regards,

Jung-Eun Kim

Academic Editor

PLOS ONE

Journal Requirements:

Reviewers' comments:

Reviewer's Responses to Questions

**Comments to the Author**

1. Is the manuscript technically sound, and do the data support the conclusions?

Reviewer #1: Partly

Reviewer #2: Yes

Reviewer #3: Yes

2. Has the statistical analysis been performed appropriately and rigorously? 

Reviewer #1: Yes

Reviewer #2: Yes

Reviewer #3: Yes

3. Have the authors made all data underlying the findings in their manuscript fully available?

Reviewer #1: Yes

Reviewer #2: Yes

Reviewer #3: Yes

4. Is the manuscript presented in an intelligible fashion and written in standard English?

Reviewer #1: Yes

Reviewer #2: Yes

Reviewer #3: Yes

5. Review Comments to the Author

Reviewer #1: Manuscript Number: PONE-D-20-37281

In this research article, Davis et al. reported a mouse model in which NF-�B inducing kinase (NIK) was constitutively activated under the control of Osx or FSP1 promoter. In these mice, soft tissue sarcoma developed with the incidence of tumors higher for the FSP1-promoter controlled condition than the Osx-promoter mediated transgenic background. The authors demonstrated that tumor cells exhibited nuclear RelB staining and that tumors were mesenchymal cells based on immune-staining of cancer tissues with aSMA or vimentin antibody. They also presented RNA-sequencing data to demonstrate that NF-kB pathway is activated in tumor tissues. The authors’ main conclusion is that constitutive NIK activation in mesenchymal cells is responsible for sarcoma development. It is certainly interesting that when NIK mutant that cannot bind to TRAF3 is expressed under the control of the FSP1-promoter, the incidence of sarcoma is quite high. However, from current data, it is unclear if unrestrained activation of NIK is indeed responsible for sarcoma formation. Empirical evidence is not robust enough to make this claim. Although NT3 transgene has been shown to become ligand-independent in transgenic mice (under the control of different promoters), current paper should provide sufficient evidence to substantiate the claim that constitutive NIK activation drives sarcoma.

First, the authors claimed that NIK is activated in tumor cells. Nevertheless, the only line of evidence for this argument is nuclear RelB in tumor tissues. RelB staining in tumor tissues only without staining data in normal tissues is not sufficient to deem NIK constitutively active in these tumors. To substantiate that NIK is indeed active in sarcoma and that it is responsible for cancer phenotype, the authors should preform additional experiments (details below). In addition, they should perform a systematic analysis of the expression level of genes that are known to be regulated by NIK using their RNA-sequencing data. Second, it is not entirely clear if NIK activity is responsible for tumor development. To make such an argument, NIK’s activity has to be disrupted or writing should be revised to accurately reflect the data. Third, assuming that NT3 is responsible for cancer development, it is not clear sarcoma development is as a result of NIK activation in cells of mesenchymal lineage or other cell types (such as immune cells). The activity of FSP1 promoter is not specific to fibroblasts. A considerable percentage of other cell types are also positive for FSP1 (Kong et al., 2013). The authors also cited this as a pitfall however, their current data did not address this concern. Please see below for detailed explanations.

Major Issues

1) Nuclear RelB suggests activation of the non-canonical NF-kB pathway. However, without evaluating RelB staining in tissues that do not express NT3 transgene, it is hard to conclude the extent to which constitutive NIK activation contributes to nuclear RelB. I understand based on the information provided by the authors that sarcoma can develop throughout the body but in some defined regions. The authors should perform RelB staining using tissues from these regions or cells positive for FSP1/Osx derived from control mice (mice with either NT3 alone or Cre alone) which do not express NT3 transgene. It is reassuring to see that stromal cells are negative for RelB in Figure 3 however, without knowing the identity of those cells it is hard to make a conclusion about nuclear RelB level in cells that do not express NT3 transgene. In addition, the authors should use at least another independent approach to confirm that the non-canonical pathway is active. For example, they could determine the processing of p100 to p52. Another suggestion is to perform a systematic analysis of expression of genes that are known to be regulated by NIK using their RNA seq to provide further evidence for high NIK activity in tumor cells.

The observation that RelB is nuclear does not exclude the possibility that the canonical NF-kB pathway is also active in NT3 genetic background. It is established that NIK can activate both canonical and alternative pathways (Staudt 2010). Thus, the possibility that both pathways may contribute to tumor development still exists. Evaluating the status of the canonical pathway activation would be helpful to address this question. The presence of nuclear RelB in tumor cells is not sufficient for an argument that tumor development is driven by the non-canonical pathway. To make such a claim, the authors should disrupt the activation of the non-canonical pathway and demonstrate reduced propensity for tumor formation or at least provide sufficient evidence that the non-canonical pathway is more active in tumor cells compared to healthy tissues along with the status for the canonical pathway. In the case where evidence is lacking, the authors should appropriately revise the arguments and include careful discussion of the canonical and non-canonical pathway contribution to tumorigenesis.

2) The authors acknowledged that FSP1 promoter is not restricted only to fibroblasts. To address this concern, they stained tumor tissues with CD45 and concluded that CD45 staining is mostly from immune infiltrates. However, this experiment alone does not resolve the issue. Mesenchymal marker staining and CD45 staining data only demonstrate that most tumor cells are of mesenchymal origin. It does not provide evidence to support the idea that cancer arose from NT3 expressing mesenchymal cells. How do we know that tumor development is a result of NT3 expression in fibroblasts but not due to the expression of NT3 in cells with hematopoietic or other lineages positive for FSP1? At a minimum, the authors should preform immunofluorescent staining to demonstrate that majority of tumor tissue is GFP+ and aSMA/vimentin+ since NT3 transgene carries GFP. Even with these data, the authors should carefully discuss the limitations of the data and possible alternative explanations for observed phenotypes.

3) The authors mentioned that they did not have a true control that they could compare to for the RNA seq data. Although I recognize the difficulty, they could have used non-cancerous cells positive for Osx or FSP1 sorted from mice or cell lines that are known to be active for these promoters. Ideally, this type of control should be used. Without a true control, analysis of the gene expression data is not robust enough to derive a confident conclusion. I appreciate their discussion on the limitations of their gene expression data.

On a related note, it is unclear how fold changes were determined for differentially expressed genes. The authors stated “we compared both Osx-Cre;NT3 and FSP1-Cre;NT3 tumors together (NT3 tumors) vs all benign tissues shown in Fig 5A to generate a list of differentially expressed genes (DEGs).” However, in the excel file that is labeled as Suppl Table 6, formulae to calculate the fold changes were not provided. For complete transparency, the authors should provide this information as part of the excel sheet and detailed explanation of how the values were derived in the method section. For example, how were “mean normal counts” determined?

4) IHC data: Quantification of the staining would be helpful to inform the reader of the variability and reproducibility of the staining. For instance, are most images 80% positive for the staining? What percentage of the tumors resembles the representative images? Quantification may be in the form of scoring of each image (low staining, medium staining or high staining) or percentage of positive cells per field (if that is feasible).

5) It is puzzling and disconcerting to observe vimentin staining mostly in the periphery of FSP1-Cre; NT3 tumors unlike the staining for a-SMA. This raises the question of whether cells with active Osx or FSP1-promoter are positive for vimentin and a-SMA. The authors should address this question using Tdt mice that they used in their study with appropriate promoters driving Cre (to examine Tdt and vimentin/a-SMA positive cells). This can help provide some answers to the question regarding the lineage of the tumor.

6) In PCA, why did Osx-Cre; NT3 bone samples group away from Osx-Cre; NT3 tumor samples? Given that the protein product of NT3 transgene is responsible for NF-kB pathway activation, they should be at least similar in some aspects. This observation suggests that considerable changes not directly influenced by NT3 transgene occurred in tumors. Can the authors provide better insights into this?

7) Due to the spontaneous nature, transgenic mouse models reported in this paper are not entirely genetically tractable. The authors recognized this aspect as well. Further characterization of genetic changes will be helpful to provide the relevance of this mouse model to human soft tissue sarcoma. This will also increase the utility of these mouse models in the field. For instance, the authors should characterize the status of some common genetic changes associated with soft tissue sarcoma (Dodd et al., 2010). The authors did make a statement that they did not observe notable changes in expression of genes whose alterations are associated with sarcoma. However, as the authors described, their differential gene expression data are limited by the lack of true controls. Thus, instead of relying on the RNA seq data, assessing the status of a few defined sarcoma-associated genetic changes would be helpful (e.g. Western blot analysis using tumor tissues and healthy tissues or cells positive for FSP1 or Osx)

References

Dodd RD, Mito JK, Kirsch DG. Animal models of soft-tissue sarcoma. Dis Model Mech. 2010 Sep-Oct;3(9-10):557-66. doi: 10.1242/dmm.005223. Epub 2010 Aug 16. PMID: 20713645; PMCID: PMC2931534.

Kong P, Christia P, Saxena A, Su Y, Frangogiannis NG. Lack of specificity of fibroblast-specific protein 1 in cardiac remodeling and fibrosis. Am J Physiol Heart Circ Physiol. 2013 Nov 1;305(9):H1363-72. doi: 10.1152/ajpheart.00395.2013. Epub 2013 Aug 30. PMID: 23997102; PMCID: PMC3840245.

Staudt LM. Oncogenic activation of NF-kappaB. Cold Spring Harb Perspect Biol. 2010 Jun;2(6):a000109. doi: 10.1101/cshperspect.a000109. Epub 2010 Apr 21. PMID: 20516126; PMCID: PMC2869521.

Reviewer #2: The submitted manuscript by Davis et al presents two new models of mesenchymal tumors. These are exciting and potentially useful new models of sarcoma. I have two recommendations: 1) the authors should include a board certified pathologist or a veterinary pathologist (preferred) for a detailed assessment of IHC staining. If they have already done so it is not clear from the methods. 2) The gene ontology analysis appears to compare the NT3 tumors (though its not clear whether these are Osx or FSP driven tumors) to ALL the normal tissues in the PCA. Both the Fsp and Osx driven tumors have the closest relationship with connective tissue, which is perfectly reasonable. I think it might be more accurate to compare the tumors to the cartilage specifically and show that data.

In general. this short report of new sarcoma models taking advantage of deregulated non-canonical NF-kB signaling has the potential to be very useful to the field.

Reviewer #3: This manuscript by Davis et al. provides a descriptive account of a novel mouse model of soft tissue sarcoma driven by Osterix or Fibroblast-Specific Protein 1 lineage cells with constitutive activation of the NF-kB pathway. The sarcoma subtype does not have a clear human counterpart, which is a limitation but not surprising in the context of a genetic engineered mouse model. The work provides a novel mouse model of sarcoma and an opportunity to expand our understanding of the NF-kB pathway in sarcomagenesis. The study is well written and I do not have major concerns in its present form. I do have the following minor suggestions/questions:

1) In the description of where tumors arise in the mice, there is a range of for each location. Please clarify what this range means as I think it would be fine to just state the sample size and the observation.

2) While the model system is highly penetrant, there is a comment made the rarity of tumors overall. Was there any thought to do a genomic analysis to assess for gene deletion/amplification? Any possible role for an environmental stimulus such as tissue injury? Is expression of the Osx or FSP higher in the face vs. other tissue types?

3) Was there any expression of myogenic markers in the tumors, ie myogenin or MyoD?

6. PLOS authors have the option to publish the peer review history of their article (what does this mean?). If published, this will include your full peer review and any attached files.

Reviewer #1: No

Reviewer #2: No

Reviewer #3: No

---

## [Author Response · Author response to Decision Letter 0]

30 Apr 2021

We thank the reviewers for their thoughtful consideration of our manuscript. Below are their specific comments and our detailed responses to each one. We believe that consideration of these issues has improved the manuscript considerably.

Reviewer #1: Manuscript Number: PONE-D-20-37281

Major Issues

1) Nuclear RelB suggests activation of the non-canonical NF-kB pathway. However, without evaluating RelB staining in tissues that do not express NT3 transgene, it is hard to conclude the extent to which constitutive NIK activation contributes to nuclear RelB. I understand based on the information provided by the authors that sarcoma can develop throughout the body but in some defined regions. The authors should perform RelB staining using tissues from these regions or cells positive for FSP1/Osx derived from control mice (mice with either NT3 alone or Cre alone) which do not express NT3 transgene. It is reassuring to see that stromal cells are negative for RelB in Figure 3 however, without knowing the identity of those cells it is hard to make a conclusion about nuclear RelB level in cells that do not express NT3 transgene. In addition, the authors should use at least another independent approach to confirm that the non-canonical pathway is active. For example, they could determine the processing of p100 to p52. Another suggestion is to perform a systematic analysis of expression of genes that are known to be regulated by NIK using their RNA seq to provide further evidence for high NIK activity in tumor cells.

Response: Our point is not that normal tissues do not have cells that activate alternative NF-kB and therefore have nuclear RelB, but that the tumors have strong and very uniform activation. Furthermore, it was not possible to examine RelB staining in tumors from Ctrl mice (non NT3 transgene setting) since Ctrl mice never developed tumors up to 1 year of age. We do not have RelB IHC in normal (non-transgenic) skin sections. However, the skin overlying FSP1-Cre;NT3 tumor 187 shows rare RelB+ cells by IHC which may be DFs that express the transgene. We have revised the RelB IHC figure (now Fig 6) to show more of the surrounding tissues for comparison, and included all 6 tumors stained. Additionally, we now show a set of 12 NF-kB regulated genes that are more highly expressed in the tumors than in our panel of normal tissues as well as in the littermate control non-transgenic bone from our prior study, according to RNA-Seq data (now Fig 5). We consider this to be very strong evidence of NF-kB activation in the tumors, in conjunction with the high levels of nuclear RelB. 

The observation that RelB is nuclear does not exclude the possibility that the canonical NF-kB pathway is also active in NT3 genetic background. It is established that NIK can activate both canonical and alternative pathways (Staudt 2010). Thus, the possibility that both pathways may contribute to tumor development still exists. Evaluating the status of the canonical pathway activation would be helpful to address this question. The presence of nuclear RelB in tumor cells is not sufficient for an argument that tumor development is driven by the non-canonical pathway. To make such a claim, the authors should disrupt the activation of the non-canonical pathway and demonstrate reduced propensity for tumor formation or at least provide sufficient evidence that the non-canonical pathway is more active in tumor cells compared to healthy tissues along with the status for the canonical pathway. In the case where evidence is lacking, the authors should appropriately revise the arguments and include careful discussion of the canonical and non-canonical pathway contribution to tumorigenesis.

Response: We do not claim that the canonical or classical pathway is not activated by the NT3 transgene, as in fact we have observed canonical events such as p65 nuclear translocation[1], [2]. We have carefully worded the manuscript, and indicate only NF-kB activation, not specifically alternative NF-kB, is driven by the NT3 transgene. Additional discussion of this point has been added, lines 607-615.

2) The authors acknowledged that FSP1 promoter is not restricted only to fibroblasts. To address this concern, they stained tumor tissues with CD45 and concluded that CD45 staining is mostly from immune infiltrates. However, this experiment alone does not resolve the issue. Mesenchymal marker staining and CD45 staining data only demonstrate that most tumor cells are of mesenchymal origin. It does not provide evidence to support the idea that cancer arose from NT3 expressing mesenchymal cells. How do we know that tumor development is a result of NT3 expression in fibroblasts but not due to the expression of NT3 in cells with hematopoietic or other lineages positive for FSP1? At a minimum, the authors should preform immunofluorescent staining to demonstrate that majority of tumor tissue is GFP+ and aSMA/vimentin+ since NT3 transgene carries GFP. Even with these data, the authors should carefully discuss the limitations of the data and possible alternative explanations for observed phenotypes.

Response: We have shown that the majority of tumor cells are GFP+ (Fig 2) and aSMA+ (Fig 3), and we have indicated in the text the extent of staining for each. Our matched images of aSMA and CD45 from adjacent sections (Fig 3) show little overlap in these populations. Based on these findings, we find it extremely unlikely that the tumors could arise from NT3+/CD45+ cells. This is discussed on lines 585-587 in the discussion. 

3) The authors mentioned that they did not have a true control that they could compare to for the RNA seq data. Although I recognize the difficulty, they could have used non-cancerous cells positive for Osx or FSP1 sorted from mice or cell lines that are known to be active for these promoters. Ideally, this type of control should be used. Without a true control, analysis of the gene expression data is not robust enough to derive a confident conclusion. I appreciate their discussion on the limitations of their gene expression data.

Response: We have directly compared the tumors to Osx-Cre;NT3 bones, and they show a similar, but more dramatic, increase in expression of NF-kB target genes in the new Fig 5B/C. Given the relative rarity of Cre+ cells in benign skin, it is not practical to isolate these cells to perform RNA-Seq.

On a related note, it is unclear how fold changes were determined for differentially expressed genes. The authors stated “we compared both Osx-Cre;NT3 and FSP1-Cre;NT3 tumors together (NT3 tumors) vs all benign tissues shown in Fig 5A to generate a list of differentially expressed genes (DEGs).” However, in the excel file that is labeled as Suppl Table 6, formulae to calculate the fold changes were not provided. For complete transparency, the authors should provide this information as part of the excel sheet and detailed explanation of how the values were derived in the method section. For example, how were “mean normal counts” determined?

Response: We have added more detailed methodology describing our calculations to the methods section (lines 238-242) as well as added a separate tab to Supplementary Table 6.

4) IHC data: Quantification of the staining would be helpful to inform the reader of the variability and reproducibility of the staining. For instance, are most images 80% positive for the staining? What percentage of the tumors resembles the representative images? Quantification may be in the form of scoring of each image (low staining, medium staining or high staining) or percentage of positive cells per field (if that is feasible).

Response: Additional description of the extent of staining has been added to the text describing IHC data in Figs 2, 3, and 6.

5) It is puzzling and disconcerting to observe vimentin staining mostly in the periphery of FSP1-Cre; NT3 tumors unlike the staining for a-SMA. This raises the question of whether cells with active Osx or FSP1-promoter are positive for vimentin and a-SMA. The authors should address this question using Tdt mice that they used in their study with appropriate promoters driving Cre (to examine Tdt and vimentin/a-SMA positive cells). This can help provide some answers to the question regarding the lineage of the tumor.

Response: We acknowledge that there are limitations to the IHC data. We previously showed that isolated tdT+ cells from Osx-Cre reporter mice also express a-SMA and Vimentin mRNA by q-RT-PCR [3]. We do not have mice with Tdt and the NT3 transgene so we cannot stain for these markers as well as Tdt, and also do not have FSP1-Cre;Tdt reporter mice available. It would take 5-6 months to generate such mice and wait for tumors to occur. However, our IHC for a-SMA and GFP on adjacent sections (from different tumors to those shown in Fig 3 clearly demonstrates a consistent pattern of overlapping staining for a-SMA and GFP (Reviewer Figure 1). We do not know why the vimentin is less uniformly expressed in the tumor cells from the FSP1-Cre line. Nevertheless, in conjunction with the fact that a diagnosis of poorly differentiated sarcoma is often one of exclusion, the totality of the evidence points towards a mesenchymal origin without differentiation towards a specific lineage.

Reviewer Figure 1: Adjacent sections of tumors (different mice than in Fig 3), stained for GFP and �SMA.

6) In PCA, why did Osx-Cre; NT3 bone samples group away from Osx-Cre; NT3 tumor samples? Given that the protein product of NT3 transgene is responsible for NF-kB pathway activation, they should be at least similar in some aspects. This observation suggests that considerable changes not directly influenced by NT3 transgene occurred in tumors. Can the authors provide better insights into this?

Response: The original PCA we showed, now Fig 4A, included all genes. In this analysis, specific pathway activation was not apparent while tissue specific expression patterns were, explaining why tumors segregated away from NT3+bone. In fact, even when we performed a gene set enrichment analysis (GSEA) of top NF-kB-related genes (approx. 500 genes) comparing only tumors and bone, tissue type seemed dominant to the transgene (new S5 Fig and new Supp Table 8). To further refine our analysis to only NT3 transgene target genes, we analyzed 12 genes found to be differentially expressed in age/sex-matched NT3 vs Ctrl bones that contained 2 or more �B response elements[2]. We found a similar pattern of upregulation between NT3 tumors and the normal tissues in our RNA-Seq panel (New Fig 5B-C), which we consider to be strong evidence of NT3 transgene activation as they were consistent with both Cre-drivers in the tumors.

7) Due to the spontaneous nature, transgenic mouse models reported in this paper are not entirely genetically tractable. The authors recognized this aspect as well. Further characterization of genetic changes will be helpful to provide the relevance of this mouse model to human soft tissue sarcoma. This will also increase the utility of these mouse models in the field. For instance, the authors should characterize the status of some common genetic changes associated with soft tissue sarcoma (Dodd et al., 2010). The authors did make a statement that they did not observe notable changes in expression of genes whose alterations are associated with sarcoma. However, as the authors described, their differential gene expression data are limited by the lack of true controls. Thus, instead of relying on the RNA seq data, assessing the status of a few defined sarcoma-associated genetic changes would be helpful (e.g. Western blot analysis using tumor tissues and healthy tissues or cells positive for FSP1 or Osx)

Response: The need for only a single transgene and 100% penetrance in a relatively short time window (tumors occur mainly between 3 and 5 months of age) makes this model easy to combine with other genes or pharmacological agents. Clearly, there is much more work that can be done to understand the nature of the tumors in our model, and the factors involved in their initiation. However, we consider this work beyond the scope of this initial report.

Reviewer #2: The submitted manuscript by Davis et al presents two new models of mesenchymal tumors. These are exciting and potentially useful new models of sarcoma. I have two recommendations: 1) the authors should include a board certified pathologist or a veterinary pathologist (preferred) for a detailed assessment of IHC staining. If they have already done so it is not clear from the methods.

Response: The senior author is a board certified pathologist with ~30 years of experience in the analysis of mouse and human tissues, now noted in methods on line 204. 

2) The gene ontology analysis appears to compare the NT3 tumors (though its not clear whether these are Osx or FSP driven tumors) to ALL the normal tissues in the PCA. Both the Fsp and Osx driven tumors have the closest relationship with connective tissue, which is perfectly reasonable. I think it might be more accurate to compare the tumors to the cartilage specifically and show that data.

Response: We have clarified the grouping of NT3 tumors in the methods on lines 247-248. Although the PCA showed the NT3 tumors to be closest to cartilage, there is neither an anatomic connection of the tumors to any cartilaginous tissue nor any histologic features of cartilaginous differentiation, and expression of cartilage marker genes is very low (new S4 Fig). Therefore, we did not perform GO analysis between tumors and cartilage. 

In general. this short report of new sarcoma models taking advantage of deregulated non-canonical NF-kB signaling has the potential to be very useful to the field.

Response: Thank you for your consideration of the value of this model.

Reviewer #3: This manuscript by Davis et al. provides a descriptive account of a novel mouse model of soft tissue sarcoma driven by Osterix or Fibroblast-Specific Protein 1 lineage cells with constitutive activation of the NF-kB pathway. The sarcoma subtype does not have a clear human counterpart, which is a limitation but not surprising in the context of a genetic engineered mouse model. The work provides a novel mouse model of sarcoma and an opportunity to expand our understanding of the NF-kB pathway in sarcomagenesis. The study is well written and I do not have major concerns in its present form. I do have the following minor suggestions/questions:

1) In the description of where tumors arise in the mice, there is a range of for each location. Please clarify what this range means as I think it would be fine to just state the sample size and the observation.

Response: The percentage range listed refers to the incidence in males vs. females, now clarified in the text.

2) While the model system is highly penetrant, there is a comment made the rarity of tumors overall. Was there any thought to do a genomic analysis to assess for gene deletion/amplification? Any possible role for an environmental stimulus such as tissue injury? Is expression of the Osx or FSP higher in the face vs. other tissue types?

Response: Although interesting, we consider analysis for gene amplification/deletion beyond the scope of the current study. We also have not directly compared Osx or FSP expression in skin on the face vs other sites. Injury and resulting inflammation, from grooming or fighting, may indeed play a role in tumorigenesis, by inducing expression of tumorigenic cofactors or enhancing NT3 transgene expression. We found BMP7 to be highly expressed in tumors, and this factor can be increased by inflammation in skin and induce Osx [4],[5]. However, in the course of our previous study on the Osx-Cre;NT3 mice we performed OVX surgeries and followed mice for 4 weeks, but did not observe any tumors at the surgical sites. Therefore, significantly more work needs to be done to resolve this issue.

3) Was there any expression of myogenic markers in the tumors, ie myogenin or MyoD?

Response: In the RNA-Seq analysis, expression of MyoD1, Myogenin, Myf5, and Desmin were all very low compared to the skeletal muscle samples, now shown in S4 Fig.

References:

[1] C. Yang, K. McCoy, J. L. Davis, M. Schmidt-Supprian, Y. Sasaki, R. Faccio, and D. V. Novack, “NIK stabilization in osteoclasts results in osteoporosis and enhanced inflammatory osteolysis.,” PLoS ONE, vol. 5, no. 11, p. e15383, 2010.

[2] J. L. Davis, L. Cox, C. Shao, C. Lyu, S. Liu, R. Aurora, and D. J. Veis, “Conditional Activation of NF‐κB Inducing Kinase (NIK) in the Osteolineage Enhances Both Basal and Loading‐Induced Bone Formation,” Journal of Bone and Mineral Research, vol. 34, no. 11, pp. 2087–2100, Aug. 2019.

[3] B. Ricci, E. Tycksen, H. Celik, J. I. Belle, F. Fontana, R. Civitelli, and R. Faccio, “Osterix-Cre marks distinct subsets of CD45- and CD45+ stromal populations in extra-skeletal tumors with pro-tumorigenic characteristics,” Elife, vol. 9, pp. 7556–29, Aug. 2020.

[4] I. Borek, R. Köffel, J. Feichtinger, M. Spies, E. Glitzner-Zeis, M. Hochgerner, T. Sconocchia, C. Krump, C. Tam-Amersdorfer, C. Passegger, T. Benezeder, J. Tittes, A. Redl, C. Painsi, G. G. Thallinger, P. Wolf, G. Stary, M. Sibilia, and H. Strobl, “BMP7 aberrantly induced in the psoriatic epidermis instructs inflammation-associated Langerhans cells.,” J Allergy Clin Immunol, vol. 145, no. 4, pp. 1194–1207.e11, Apr. 2020.

[5] T. Sconocchia, M. Hochgerner, E. Schwarzenberger, C. Tam-Amersdorfer, I. Borek, T. Benezeder, T. Bauer, V. Zyulina, C. Painsi, C. Passegger, P. Wolf, M. Sibilia, and H. Strobl, “Bone morphogenetic protein signaling regulates skin inflammation via modulating dendritic cell function.,” J Allergy Clin Immunol, Oct. 2020.

---

## [Decision Letter · Decision Letter 1]

17 May 2021

PONE-D-20-37281R1

Constitutive Activation of NF-κB Inducing Kinase (NIK) in the Mesenchymal Lineage using Osterix (Sp7)- or Fibroblast-Specific Protein 1 (S100a4)-Cre Drives Spontaneous Soft Tissue Sarcoma

PLOS ONE

Dear Dr. Veis,

Thank you for submitting your manuscript to PLOS ONE. After careful consideration, we feel that it has merit but does not fully meet PLOS ONE’s publication criteria as it currently stands. Therefore, we invite you to submit a revised version of the manuscript that addresses the points raised during the review process.

Please find comments raised by reviewer #1 and respond sincerely.

We look forward to receiving your revised manuscript.

Kind regards,

Jung-Eun Kim

Academic Editor

PLOS ONE

Journal Requirements:

Reviewers' comments:

Reviewer's Responses to Questions

**Comments to the Author**

1. If the authors have adequately addressed your comments raised in a previous round of review and you feel that this manuscript is now acceptable for publication, you may indicate that here to bypass the “Comments to the Author” section, enter your conflict of interest statement in the “Confidential to Editor” section, and submit your "Accept" recommendation.

Reviewer #1: All comments have been addressed

Reviewer #2: All comments have been addressed

2. Is the manuscript technically sound, and do the data support the conclusions?

Reviewer #1: Yes

Reviewer #2: Yes

3. Has the statistical analysis been performed appropriately and rigorously? 

Reviewer #1: Yes

Reviewer #2: Yes

4. Have the authors made all data underlying the findings in their manuscript fully available?

Reviewer #1: Yes

Reviewer #2: Yes

5. Is the manuscript presented in an intelligible fashion and written in standard English?

Reviewer #1: Yes

Reviewer #2: Yes

6. Review Comments to the Author

Reviewer #1: Responses to the authors

Manuscript Number: PONE-D-20-37281

Major Issues

1) Nuclear RelB suggests activation of the non-canonical NF-kB pathway. However, without evaluating RelB staining in tissues that do not express NT3 transgene, it is hard to conclude the extent to which constitutive NIK activation contributes to nuclear RelB. I understand based on the information provided by the authors that sarcoma can develop throughout the body but in some defined regions. The authors should perform RelB staining using tissues from these regions or cells positive for FSP1/Osx derived from control mice (mice with either NT3 alone or Cre alone) which do not express NT3 transgene. It is reassuring to see that stromal cells are negative for RelB in Figure 3 however, without knowing the identity of those cells it is hard to make a

conclusion about nuclear RelB level in cells that do not express NT3 transgene. In addition, the authors should use at least another independent approach to confirm that the non-canonical pathway is active. For example, they could determine the processing of p100 to p52. Another suggestion is to perform a systematic analysis of expression of genes that are known to be regulated by NIK using their RNA seq to provide further evidence for high NIK activity in tumor cells.

Response: Our point is not that normal tissues do not have cells that activate alternative NF-kB and therefore have nuclear RelB, but that the tumors have strong and very uniform activation. Furthermore, it was not possible to examine RelB staining in tumors from Ctrl mice (non NT3 transgene setting) since Ctrl mice never developed tumors up to 1 year of age. We do not have RelB IHC in normal (non-transgenic) skin sections. However, the skin overlying FSP1-Cre;NT3 tumor 187 shows rare RelB+ cells by IHC which may be DFs that express the transgene. We have revised the RelB IHC figure (now Fig 6) to show more of the surrounding tissues for comparison, and included all 6 tumors stained. Additionally, we now show a set of 12 NF-kB

regulated genes that are more highly expressed in the tumors than in our panel of normal tissues as well as in the littermate control non-transgenic bone from our prior study, according to RNA-Seq data (now Fig 5). We consider this to be very strong evidence of NF-kB activation in the tumors, in conjunction with the high levels of nuclear RelB.

The reviewer did not suggest tumors from Ctrl mice to be stained with RelB. The suggestion was to perform “RelB staining using tissues from these regions or cells positive for FSP1/Osx derived from control mice (mice with either NT3 alone or Cre alone) which do not express NT3 transgene”. The reviewer understands that mice with NT3 alone or Cre are simply controls in which NIK is not constitutively active and that transgene-driven tumors do not develop in these mice. Therefore, the reviewer understands that the isolation of healthy tissues from these mice does not require any waiting time and that these control mice are already in existence. Hence, the argument about tumor development taking up to 1 year of age confuses the reviewer.

Including the surrounding tissues help to ensure that RelB staining is specific to tumors. Independent assessment of NF-kB regulated gene expression in tumor tissues compared to the tissues without the transgene also supports the argument that NF-kB pathway is activated.

The observation that RelB is nuclear does not exclude the possibility that the canonical NF-kB pathway is also active in NT3 genetic background. It is established that NIK can activate both canonical and alternative pathways (Staudt 2010). Thus, the possibility that both pathways may contribute to tumor development still exists. Evaluating the status of the canonical pathway activation would be helpful to address this question. The presence of nuclear RelB in tumor cells is not sufficient for an argument that tumor development is driven by the non-canonical pathway. To make such a claim, the authors should disrupt the activation of the noncanonical pathway and demonstrate reduced propensity for tumor formation or at least provide sufficient

evidence that the non-canonical pathway is more active in tumor cells compared to healthy tissues along with the status for the canonical pathway. In the case where evidence is lacking, the authors should appropriately revise the arguments and include careful discussion of the canonical and non-canonical pathway contribution to tumorigenesis.

Response: We do not claim that the canonical or classical pathway is not activated by the NT3 transgene, as in fact we have observed canonical events such as p65 nuclear translocation[1], [2]. We have carefully worded the manuscript, and indicate only NF-kB activation, not specifically alternative NF-kB, is driven by the NT3 transgene. Additional discussion of this point has been added, lines 607-615.

Revisions to the manuscript to reflect these points have been noted.

2) The authors acknowledged that FSP1 promoter is not restricted only to fibroblasts. To address this concern, they stained tumor tissues with CD45 and concluded that CD45 staining is mostly from immune infiltrates. However, this experiment alone does not resolve the issue. Mesenchymal marker staining and CD45 staining data only demonstrate that most tumor cells are of mesenchymal origin. It does not provide evidence to support the idea that cancer arose from NT3 expressing mesenchymal cells. How do we know that tumor development is a result of NT3 expression in fibroblasts but not due to the expression of NT3 in cells with hematopoietic or

other lineages positive for FSP1? At a minimum, the authors should preform immunofluorescent staining to demonstrate that majority of tumor tissue is GFP+ and aSMA/vimentin+ since NT3 transgene carries GFP. Even with these data, the authors should carefully discuss the limitations of the data and possible alternative explanations for observed phenotypes.

Response: We have shown that the majority of tumor cells are GFP+ (Fig 2) and aSMA+ (Fig 3), and we have indicated in the text the extent of staining for each. Our matched images of aSMA and CD45 from adjacent sections (Fig 3) show little overlap in these populations. Based on these findings, we find it extremely unlikely that the tumors could arise from NT3+/CD45+ cells. This is discussed on lines 585-587 in the discussion.

Changes have been noted.

3) The authors mentioned that they did not have a true control that they could compare to for the RNA seq data. Although I recognize the difficulty, they could have used non-cancerous cells positive for Osx or FSP1 sorted from mice or cell lines that are known to be active for these promoters. Ideally, this type of control should be used. Without a true control, analysis of the gene expression data is not robust enough to derive a confident conclusion. I appreciate their discussion on the limitations of their gene expression data.

Response: We have directly compared the tumors to Osx-Cre;NT3 bones, and they show a similar, but more dramatic, increase in expression of NF-kB target genes in the new Fig 5B/C. Given the relative rarity of Cre+ cells in benign skin, it is not practical to isolate these cells to perform RNA-Seq.

The suggestion was to isolate non-cancerous cells positive for Osx or FSP1. For example, the authors reported that FSP-1 “predominantly shows expression in the fibroblast population of multiple organs but has also been documented in the myeloid lineage[31], [32], [43]-[45].” While it is understandable that the authors may not want to perform RNA seq experiments due to the time constraint, their response on “the relative rarity of Cre+ cells in benign skin” confuses the reviewer.

On a related note, it is unclear how fold changes were determined for differentially expressed genes. The authors stated “we compared both Osx-Cre;NT3 and FSP1-Cre;NT3 tumors together (NT3 tumors) vs all benign tissues shown in Fig 5A to generate a list of differentially expressed genes (DEGs).” However, in the excel file that is labeled as Suppl Table 6, formulae to calculate the fold changes were not provided. For complete transparency, the authors should provide this information as part of the excel sheet and detailed explanation of how the values were derived in the method section. For example, how were “mean normal counts” determined?

Response: We have added more detailed methodology describing our calculations to the methods section (lines 238-242) as well as added a separate tab to Supplementary Table 6.

Changes have been noted. The reviewer appreciates that the authors referenced another paper for data analysis.

4) IHC data: Quantification of the staining would be helpful to inform the reader of the variability and reproducibility of the staining. For instance, are most images 80% positive for the staining? What percentage of the tumors resembles the representative images? Quantification may be in the form of scoring of each image (low staining, medium staining or high staining) or percentage of positive cells per field (if that is feasible).

Response: Additional description of the extent of staining has been added to the text describing IHC data in Figs 2, 3, and 6.

The descriptions added for those figures do not address the issue of reproducibility and variability. They simply describe what the reader can see in the images presented.

5) It is puzzling and disconcerting to observe vimentin staining mostly in the periphery of FSP1-Cre; NT3 tumors unlike the staining for a-SMA. This raises the question of whether cells with active Osx or FSP1-promoter are positive for vimentin and a-SMA. The authors should address this question using Tdt mice that they used in their study with appropriate promoters driving Cre (to examine Tdt and vimentin/a-SMA positive cells). This can help provide some answers to the question regarding the lineage of the tumor.

Response: We acknowledge that there are limitations to the IHC data. We previously showed that isolated tdT+ cells from Osx-Cre reporter mice also express a-SMA and Vimentin mRNA by q-RT-PCR [3]. We do not have mice with Tdt and the NT3 transgene so we cannot stain for these markers as well as Tdt, and also do not have FSP1-Cre;Tdt reporter mice available. It would take 5-6 months to generate such mice and wait for tumors to occur. However, our IHC for a-SMA and GFP on adjacent sections (from different tumors to those shown in Fig 3 clearly demonstrates a consistent pattern of overlapping staining for a-SMA and GFP (Reviewer Figure 1). We do not know why the vimentin is less uniformly expressed in the tumor cells from the FSP1-Cre line. Nevertheless, in conjunction with the fact that a diagnosis of poorly differentiated sarcoma is often one of exclusion, the totality of the evidence points towards a mesenchymal origin without differentiation towards a specific lineage.

The reviewer appreciates the explanation and the figure provided. The suggestion is to describe the limitation of vimentin staining under Discussion as well (line 572-573).

6) In PCA, why did Osx-Cre; NT3 bone samples group away from Osx-Cre; NT3 tumor samples? Given that the protein product of NT3 transgene is responsible for NF-kB pathway activation, they should be at least similar in some aspects. This observation suggests that considerable changes not directly influenced by NT3 transgene occurred in tumors. Can the authors provide better insights into this?

Response: The original PCA we showed, now Fig 4A, included all genes. In this analysis, specific pathway activation was not apparent while tissue specific expression patterns were, explaining why tumors segregated away from NT3+bone. In fact, even when we performed a gene set enrichment analysis (GSEA) of top NF-kBrelated genes (approx. 500 genes) comparing only tumors and bone, tissue type seemed dominant to the transgene (new S5 Fig and new Supp Table 8). To further refine our analysis to only NT3 transgene target genes, we analyzed 12 genes found to be differentially expressed in age/sex-matched NT3 vs Ctrl bones that contained 2 or more �B response elements[2]. We found a similar pattern of upregulation between NT3 tumors and the normal tissues in our RNA-Seq panel (New Fig 5B-C), which we consider to be strong evidence of NT3 transgene activation as they were consistent with both Cre-drivers in the tumors.

Including these explanations helps to understand the gene expression data better.

7) Due to the spontaneous nature, transgenic mouse models reported in this paper are not entirely genetically tractable. The authors recognized this aspect as well. Further characterization of genetic changes will be helpful to provide the relevance of this mouse model to human soft tissue sarcoma. This will also increase the utility of these mouse models in the field. For instance, the authors should characterize the status of some common genetic changes associated with soft tissue sarcoma (Dodd et al., 2010). The authors did make a statement that they did not observe notable changes in expression of genes whose alterations are associated

with sarcoma. However, as the authors described, their differential gene expression data are limited by the lack of true controls. Thus, instead of relying on the RNA seq data, assessing the status of a few defined sarcoma associated genetic changes would be helpful (e.g. Western blot analysis using tumor tissues and healthy tissues or cells positive for FSP1 or Osx)

Response: The need for only a single transgene and 100% penetrance in a relatively short time window (tumors occur mainly between 3 and 5 months of age) makes this model easy to combine with other genes or pharmacological agents. Clearly, there is much more work that can be done to understand the nature of the tumors in our model, and the factors involved in their initiation. However, we consider this work beyond the scope of this initial report.

The reviewer acknowledges that this work may be beyond the scope of this manuscript.

Reviewer #2: The authors have addressed my concerns and I have no other comments. The manuscript is appropriate for publication

7. PLOS authors have the option to publish the peer review history of their article (what does this mean?). If published, this will include your full peer review and any attached files.

Reviewer #1: No

Reviewer #2: No

---

## [Author Response · Author response to Decision Letter 1]

24 Jun 2021

We thank the reviewers and editors of their careful consideration of our revisions. We have addressed each of the remaining issues raised by Reviewer 1, as explained below. For clarity, we have copied the previous comments and our original responses (labeled #1), as well as our new responses (labeled #2), and have omitted those considered resolved on the last round of review.

Reviewer #1: Responses to the authors

Manuscript Number: PONE-D-20-37281

Major Issues

1) Critique #1: Nuclear RelB suggests activation of the non-canonical NF-kB pathway. However, without evaluating RelB staining in tissues that do not express NT3 transgene, it is hard to conclude the extent to which constitutive NIK activation contributes to nuclear RelB. I understand based on the information provided by the authors that sarcoma can develop throughout the body but in some defined regions. The authors should perform RelB staining using tissues from these regions or cells positive for FSP1/Osx derived from control mice (mice with either NT3 alone or Cre alone) which do not express NT3 transgene. It is reassuring to see that stromal cells are negative for RelB in Figure 3 however, without knowing the identity of those cells it is hard to make a conclusion about nuclear RelB level in cells that do not express NT3 transgene. In addition, the authors should use at least another independent approach to confirm that the non-canonical pathway is active. For example, they could determine the processing of p100 to p52. Another suggestion is to perform a systematic analysis of expression of genes that are known to be regulated by NIK using their RNA seq to provide further evidence for high NIK activity in tumor cells.

Response #1: Our point is not that normal tissues do not have cells that activate alternative NF-kB and therefore have nuclear RelB, but that the tumors have strong and very uniform activation. Furthermore, it was not possible to examine RelB staining in tumors from Ctrl mice (non NT3 transgene setting) since Ctrl mice never developed tumors up to 1 year of age. We do not have RelB IHC in normal (non-transgenic) skin sections. However, the skin overlying FSP1-Cre;NT3 tumor 187 shows rare RelB+ cells by IHC which may be DFs that express the transgene. We have revised the RelB IHC figure (now Fig 6) to show more of the surrounding tissues for comparison, and included all 6 tumors stained. Additionally, we now show a set of 12 NF-kB regulated genes that are more highly expressed in the tumors than in our panel of normal tissues as well as in the littermate control non-transgenic bone from our prior study, according to RNA-Seq data (now Fig 5). We consider this to be very strong evidence of NF-kB activation in the tumors, in conjunction with the high levels of nuclear RelB.

Critique #2: The reviewer did not suggest tumors from Ctrl mice to be stained with RelB. The suggestion was to perform “RelB staining using tissues from these regions or cells positive for FSP1/Osx derived from control mice (mice with either NT3 alone or Cre alone) which do not express NT3 transgene”. The reviewer understands that mice with NT3 alone or Cre are simply controls in which NIK is not constitutively active and that transgene-driven tumors do not develop in these mice. Therefore, the reviewer understands that the isolation of healthy tissues from these mice does not require any waiting time and that these control mice are already in existence. Hence, the argument about tumor development taking up to 1 year of age confuses the reviewer.

Including the surrounding tissues help to ensure that RelB staining is specific to tumors. Independent assessment of NF-kB regulated gene expression in tumor tissues compared to the tissues without the transgene also supports the argument that NF-kB pathway is activated.

Response #2: We apologize for any misinterpretation of the reviewer’s previous comment. The intent of suggesting the aging of Cre-negative littermates to 1 year was to rule out a spontaneous mutation in our colony outside of the NT3 locus and independent of Cre-mediated effects. However, we minimized founder effects by using multiple breeding pairs and routinely backcrossing both NT3 and Osx-Cre lines to C57Bl/6 mice from JAX. Additionally, the tumor penetrance is 100%, so another driver mutation is highly unlikely. 

We do not believe that performing IHC for RelB on non-transgenic skin would be a fruitful endeavor. Our images of NT3-transgenic skin near tumors and of the Osx-Cre reporter mice show very few positive cells in the dermis. A previous report by Hinata et al (ref 57, Fig 1) showed only rare cells in the dermis with NF-kB reporter activation. Therefore, in non-transgenic skin, background RelB expression, if present, would likely be even more rare than in tumor-adjacent NT3-transgenic tissue. Without co-staining for CD45 and fibroblast markers (at a minimum), it would be impossible to identify rare RelB-positive dermal cells, and thus, understand their relevance to tumorigenesis. 

2) Critique #1: The authors mentioned that they did not have a true control that they could compare to for the RNA seq data. Although I recognize the difficulty, they could have used non-cancerous cells positive for Osx or FSP1 sorted from mice or cell lines that are known to be active for these promoters. Ideally, this type of control should be used. Without a true control, analysis of the gene expression data is not robust enough to derive a confident conclusion. I appreciate their discussion on the limitations of their gene expression data.

Response#1: We have directly compared the tumors to Osx-Cre;NT3 bones, and they show a similar, but more dramatic, increase in expression of NF-kB target genes in the new Fig 5B/C. Given the relative rarity of Cre+ cells in benign skin, it is not practical to isolate these cells to perform RNA-Seq.

Critique #2: The suggestion was to isolate non-cancerous cells positive for Osx or FSP1. For example, the authors reported that FSP-1 “predominantly shows expression in the fibroblast population of multiple organs but has also been documented in the myeloid lineage[31], [32], [43]-[45].” While it is understandable that the authors may not want to perform RNA seq experiments due to the time constraint, their response on “the relative rarity of Cre+ cells in benign skin” confuses the reviewer.

Response #2: It is not clear to us which population of Osx- or FSP1-positive cells this reviewer suggests for analysis. To us, the most relevant would be cells from benign skin, which are relatively rare in the reporter mice. We believe we have included sufficient data, presented with clear indications about its limitations, to warrant publication without further experimentation.

3) Critique #1: IHC data: Quantification of the staining would be helpful to inform the reader of the variability and reproducibility of the staining. For instance, are most images 80% positive for the staining? What percentage of the tumors resembles the representative images? Quantification may be in the form of scoring of each image (low staining, medium staining or high staining) or percentage of positive cells per field (if that is feasible).

Response #1: Additional description of the extent of staining has been added to the text describing IHC data in Figs 2, 3, and 6.

Critique #2: The descriptions added for those figures do not address the issue of reproducibility and variability. They simply describe what the reader can see in the images presented. 

Response 2: We used ranges of positivity in the text along with indications of the strength (or range thereof), indicative of the variability we saw across the 6 separate tumors studied, now explicitly stated as (n=3 of each) in the text (line 410). All of the immunostaining was performed at the same time, so that it would be comparable across samples. If the journal allows a large file format, we would be happy to add scanned whole slide images for all of the IHC as supplemental information. Otherwise, we would certainly provide access to the images upon request. Unfortunately, there are no public databases in which to deposit staining data.

3) Critique #1: It is puzzling and disconcerting to observe vimentin staining mostly in the periphery of FSP1-Cre; NT3 tumors unlike the staining for a-SMA. This raises the question of whether cells with active Osx or FSP1-promoter are positive for vimentin and a-SMA. The authors should address this question using Tdt mice that they used in their study with appropriate promoters driving Cre (to examine Tdt and vimentin/a-SMA positive cells). This can help provide some answers to the question regarding the lineage of the tumor.

Response #1: We acknowledge that there are limitations to the IHC data. We previously showed that isolated tdT+ cells from Osx-Cre reporter mice also express a-SMA and Vimentin mRNA by q-RT-PCR [3]. We do not have mice with Tdt and the NT3 transgene so we cannot stain for these markers as well as Tdt, and also do not have FSP1-Cre;Tdt reporter mice available. It would take 5-6 months to generate such mice and wait for tumors to occur. However, our IHC for a-SMA and GFP on adjacent sections (from different tumors to those shown in Fig 3) clearly demonstrates a consistent pattern of overlapping staining for a-SMA and GFP (Reviewer Figure 1). We do not know why the vimentin is less uniformly expressed in the tumor cells from the FSP1-Cre line. Nevertheless, in conjunction with the fact that a diagnosis of poorly differentiated sarcoma is often one of exclusion, the totality of the evidence points towards a mesenchymal origin without differentiation towards a specific lineage.

Critique #2: The reviewer appreciates the explanation and the figure provided. The suggestion is to describe the limitation of vimentin staining under Discussion as well (line 572-573).

Response #2: Discussion has been amended to note variability of vimentin staining (lines 572-575).

---

## [Editor Report · Decision Letter 2]

28 Jun 2021

Constitutive Activation of NF-κB Inducing Kinase (NIK) in the Mesenchymal Lineage using Osterix (Sp7)- or Fibroblast-Specific Protein 1 (S100a4)-Cre Drives Spontaneous Soft Tissue Sarcoma

PONE-D-20-37281R2

Dear Dr. Veis,

We’re pleased to inform you that your manuscript has been judged scientifically suitable for publication and will be formally accepted for publication once it meets all outstanding technical requirements.

Kind regards,

Jung-Eun Kim

Academic Editor

PLOS ONE
---

## [Editor Report · Acceptance letter]

13 Jul 2021

PONE-D-20-37281R2 

Constitutive activation of NF-κB Inducing Kinase (NIK) in the mesenchymal lineage using Osterix (Sp7)- or Fibroblast-Specific Protein 1 (S100a4)-Cre drives spontaneous soft tissue sarcoma 

Dear Dr. Veis:

I'm pleased to inform you that your manuscript has been deemed suitable for publication in PLOS ONE. Congratulations! Your manuscript is now with our production department. 

Kind regards, 

on behalf of

Dr Jung-Eun Kim 

Academic Editor

PLOS ONE